# SCALE-VLP: Soft-Weighted ContrAstive VoLumEtric Vision–Language Pre-training with Spatial-Knowledge Semantics

## ABSTRACT

Vision–language models (VLMs) have demonstrated strong cross-modal capabilities, yet most work remains limited to 2D data and assumes binary supervision (*i.e.*, positive vs. negative pairs), overlooking the continuous and structured dependencies present in volumetric data such as CT. Existing approaches often treat volumetric scans as independent 2D slices, compromising spatial coherence and underutilizing rich clinical semantics. We propose **SCALE-VLP**, a soft-weighted contrastive vision-language pre-training framework that integrates (i) *volumetric spatial semantics* to preserve anatomical structure and (ii) *domain-aware, knowledge-infused semantics* (*e.g.*, radiological ontologies) to guide alignment. This yields structurally consistent and semantically grounded representations under limited supervision, demonstrating strong cross-task transferability (retrieval, report generation, and classification), and cross-domain generalizability with consistent gains without further fine-tuning. In particular, compared to the previous state of the art, SCALE-VLP achieves up to 4.3× higher top-1 CT–report retrieval, improves abnormality classification by 10 points, and reaches ROUGE-L 0.44 and BERT-F1 0.89 for report generation. Further, in zero-shot evaluation on an out-of-domain external dataset, we observe consistent gains, indicating the cross-task and cross-domain generalization ability of SCALE-VLP. [1]

## 1 INTRODUCTION

Medical imaging is crucial to modern healthcare, aiding in diagnosis, treatment planning, and monitoring. Recent advancements in AI, high-resolution imaging, and visualization techniques have significantly enhanced our ability to extract detailed clinical insights from complex medical data Perera et al. (2024); Li et al. (2025). Despite this progress, most current research still focuses on two-dimensional (2D) imaging Li et al. (2023b); Jing et al. (2020); Wu et al. (2024), *e.g.* chest X-rays Tanida et al. (2023), overlooking the increasing availability and importance of three-dimensional (3D) scans (*e.g.* computed tomography (CT) and magnetic resonance imaging (MRI)), which offer volumetric representations that are essential for capturing the full spatial complexity of anatomical structures and disease patterns. Therefore, advancing 3D medical imaging analysis constitutes a research priority with direct clinical impact.

Vision-language models (VLMs) have shown promising progress in this area by aligning visual inputs with natural language supervision to learn transferable representations. However, the inherent characteristics of volumetric data and their radiology reports pose three fundamental challenges for vision–language alignment in medical VLMs:

*(1) Data-Scarce Representation Learning*: Effective vision–language alignment in VLMs requires large-scale paired data, as exemplified by CLIP Radford et al. (2021) with 400M image–text pairs. Medical imaging, however, is constrained by privacy and diagnostic complexity; for instance, CT-RATE Hamamci et al. (2024b) provides only 24,128 CT–report pairs. Medical VLMs like M3D Bai et al. (2024), CT-CLIP Hamamci et al. (2024b), and Merlin Blankemeier et al. (2024), adopt CLIP-style training with strict one-to-one alignment, inducing binary similarity targets that underweight partial, clinically meaningful matches across studies.

---

[1] The code and analysis scripts will be released upon acceptance.

*(2) Volumetric spatial coherence modeling*: A central bottleneck lies in learning effective 3D representations that preserve the intrinsic spatial and semantic structure of the data. Unlike 2D images, 3D scans, like CT and MRI, encode fine-grained spatial details across multiple slices, making it difficult to pair them effectively with their radiology reports, including findings and impressions Lin et al. (2024); Hamamci et al. (2024a). For example, fVLM Shui et al. (2025) applies a fine-grained anatomy-level matching scheme, yet still operates on a discrete level, leaving the continuous structure of volumetric data under exploited. Recent methods, such as T3D Liu et al. (2023), attempt to address this via multi-view consistency and 3D-aware contrastive learning, but they often struggle to preserve spatial coherence across slices, which can weaken alignment with fine-grained report cues and disrupt global anatomical understanding Wang et al. (2024); Liu et al. (2024).

*(3) Medical knowledge understanding*: Radiology reports contain complex terminology, semantic relationships, and implicit references that challenge standard contrastive learning. For instance, generic models treat "consolidation" and "infiltrate" as unrelated, ignoring their synonymity in pneumonia diagnosis. Recent works such as MedKLIP Wu et al. (2023) and KAD Zhang et al. (2023) use medical ontologies (*e.g.*, UMLS Bodenreider (2004), RadGraph Jain et al.) to enhance zero-shot classification and grounding, underscoring the need for clinically informed alignment. In addition, binary contrastive pairs overlook partial relevance; for example, a multi-finding report may only partially correspond to a CT scan, leaving clinical knowledge underexploited.

To address these challenges, we propose the **SCALE-VLP** with a *soft-weighted contrastive pre-training* framework that injects *volumetric spatial coherence* and *medical knowledge* into the vision-language alignment objective. Instead of relying on binary pairwise targets, SCALE-VLP constructs a dense similarity matrix whose weights reflect semantic affinity between spatial proximity and clinical concepts within 3D scans. Volumetric structure is encoded through a volumetric kernel, while medical knowledge is infused via domain-specific embeddings. These weights rescale positive and negative terms in a CLIP/InfoNCE-style loss, encouraging the model to favor anatomically consistent and clinically plausible correspondences *without requiring additional supervision*. To summarize the contributions of this paper in three folds:

- We develop a novel Soft-Weighted Contrastive Alignment (SWCA) objective that explicitly encodes continuous, semantics-aware distances between volumetric CT data and reports, improving sample efficiency under limited supervision.
- We design a joint spatial-knowledge semantics aware alignment mechanism that constructs dense similarity matrices via volumetric spatial coherence encoding and medical knowledge fusion, enhancing the intrinsic radiological alignment between CT scans and diagnostic reports.
- We demonstrate through comprehensive experiments that SCALE-VLP exhibits strong cross-task transferability spanning CT-report retrieval, report generation, and CT abnormality classification, and cross-domain generalization to an external CT benchmark with consistent gains without further fine-tuning.

## 2 RELATED WORK

**Multi-Modal Objectives.** Multi-modal learning aims to leverage complementary cues from different modalities, such as visual and textual data, to construct more expressive feature representations Radford et al. (2021); Alayrac et al. (2022). While contrastive approaches are prevalent for aligning paired samples across domains, traditional methods often rely on softmax-based losses that require normalization over the entire batch, which introduces inefficiencies and limits scalability. To address this, SigLIP Zhai et al. (2023) proposes a more efficient sigmoid-based pairwise objective that avoids global normalization. In parallel, Srinivasa et al. Srinivasa et al. (2023) show that incorporating unimodal similarity signals can improve cross-modal alignment, especially when one modality is weaker. Additionally, Subramanian et al. Subramanian et al. (2025) highlight the benefit of using domain-specific language models as semantic priors to enhance visual tasks. Building on these advances, our method introduces a customized multi-modal learning objective that transfers medical knowledge as semantic guidance to improve clinical relevance and generalizability.

**3D medical Contrastive Learning.** Early medical multimodal contrastive learning focused on 2D image–text alignment using X-rays and paired reports, with models such as GLoRIA Huang et al. (2021), ConVIRT Zhang et al. (2022), and BioViL Boecking et al. (2022). These approaches en-

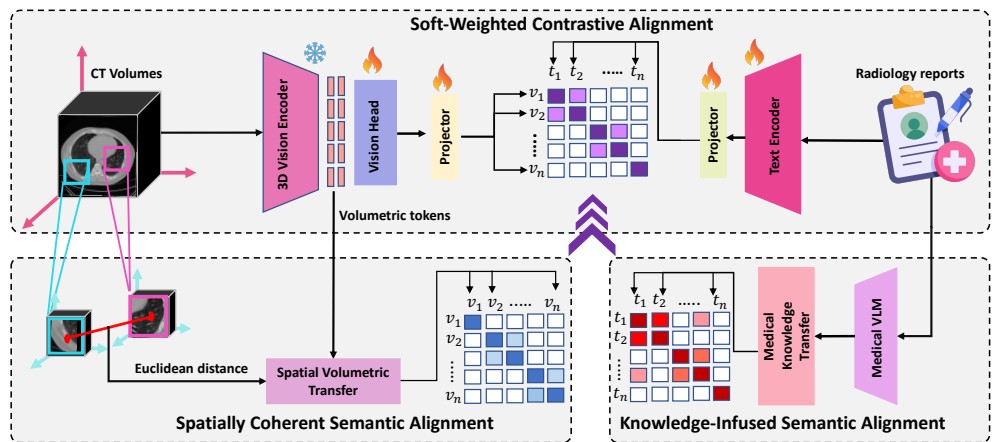

Figure 1: **SCALE-VLP framework.** A 3D vision encoder embeds CT volumes and a clinical-text encoder embeds reports. Soft-Weighted Contrastive Alignment aligns modalities using feature similarity, spatial proximity, and medical knowledge priors.

abled strong zero-shot performance in classification, retrieval, and grounding. The introduction of large-scale paired 3D datasets, such as CT-RATE Hamamci et al. (2024a) and BIMCV-R Chen et al. (2024b), has accelerated the development of volumetric contrastive models. CT-CLIP Hamamci et al. (2024a) aligns 3D CT volumes with radiology reports for zero-shot abnormality detection and retrieval. MedFinder Chen et al. (2024b) augments this with view consistency and cross-attention mechanisms. M3D Bai et al. (2024) integrates contrastive pretraining with instruction tuning to support diverse 3D tasks. However, current frameworks fall short in considering spatially grounded and clinically guided semantic alignment, which are both crucial and can lead to improvment in downstream utility in tasks like report generation, text or CT volume retrieval, or abnormality detection.

## 3 SCALE-VLP

Our goal is to learn anatomically and semantically grounded 3D representations of CT scans and align them with reports to generalize across diverse medical imaging tasks. As illustrated in Figure 1, we represent volumetric CT studies as sequences of spatially aware patch embeddings, preserving 3D structure. Each CT scan is paired with its radiology report, and both modalities are encoded using a 3D Vision Transformer and a medical-language encoder resepctively. We introduce a novel pre-training objective based on a continuously weighted similarity matrix (*Soft-Weighted Contrastive Alignment*), which softly aligns visual and textual features by incorporating volumetric spatial information (*Spatially-Guided Semantic Alignment*) and ontological medical knowledge (*Knowledge-Infused Semantic Alignment*). Once pre-trained, the learned representations can be adapted to downstream tasks such as cross-modal retrieval, abnormality classification, and report generation, demonstrating the flexibility and clinical utility of SCALE-VLP.

### 3.1 MODEL ARCHITECTURE

As shown in Figure 1, SCALE-VLP adopts a dual-encoder design that encodes volumetric CT scans using a frozen 3D ViT Dosovitskiy et al. (2021) and processes radiology reports with a fine-tuned BERT Devlin et al. (2019) encoder. The 3D ViT, pretrained on RadImageNet Mei et al. (2022), extracts patch-level features from input volumes. These tokens are passed through a lightweight Transformer layer, referred to as the vision head, which aggregates spatial information and outputs sequence-level embeddings. The resulting features are projected into a shared embedding space using a linear projection layer. The CLS token output is similarly projected and normalized to align with the vision embeddings. On the language side, reports are encoded using BioClinicalBERT Alsentzer et al. (2019), with all parameters fine-tuned during training. We compute cross-modal similarity via scaled cosine similarity, where the scaling factor is a learnable parameter initialized as in CLIP. During training, only the projectors, the vision head, and the language encoder are updated.

## 3.2 Soft-Weighted Contrastive Alignment (SWCA)

Similarity between images and text often exhibits a continuous and non-binary nature, particularly in domains such as medical imaging. For example, a CT radiology report may partially correspond to multiple scans with related pathologies, and sentences across different reports may describe the same anatomical finding from varying diagnostic perspectives. As a result, the alignment between CT volumes and reports is inherently partial and better represented along a graded spectrum. To capture this nuance, we introduce *Soft-Weighted Contrastive Alignment (SWCA)*, a novel contrastive learning objective that integrates continuous similarity with volumetric and semantic priors. Prior methods either rely on dense batch-wise normalization, which requires constructing a full similarity matrix with quadratic memory cost, or depend solely on independent pairwise scoring Radford et al. (2021); Srinivasa et al. (2023); Zhai et al. (2023). SWCA instead incorporates a soft-weighting mechanism that encodes three complementary signals: (i) graded cross-modal relevance, (ii) spatial coherence derived from 3D CT structure, and (iii) knowledge-aware priors from large medical language models. In addition, SWCA's alignment objective is formulated as a pairwise loss, eliminating collective operations while maintaining continuous alignment fidelity.

**Intra-modal similarity weight.** Given a mini-batch of $B$ paired volumetric and textual samples, we compute intra-modal encoder embeddings $\mathbf{z}_i \in \mathbb{R}^D$ for the i-th patient within each modality to estimate soft similarity-based weights. Intra-modal similarity between embeddings is computed as:

$$a_{ij} = \exp\left(\beta \cos(\mathbf{z}_i, \mathbf{z}_j)\right), \qquad a_{ii} = 0. \tag{1}$$

Row-normalization produces the similarity soft weights as follows:

$$w_{ij}^{\text{Intra-sim}} = \frac{a_{ij}}{\displaystyle\sum_{k=1,\, k\neq i}^{B} a_{ik} + \varepsilon}, \qquad w_{ii}^{\text{Intra-sim}} = 0. \tag{2}$$

where $\varepsilon$ is a small constant for numerical stability.

**Cross-modal similarity.** We extract volumetric and textual embeddings $\mathbf{v}_i, \mathbf{t}_j \in \mathbb{R}^D$ for each CT–report pair $(i, j)$. For *scaled cosine* similarity, we $\ell_2$-normalize both modalities:

$$\hat{\mathbf{v}}_i = \frac{\mathbf{v}_i}{\|\mathbf{v}_i\|_2}, \qquad \hat{\mathbf{t}}_j = \frac{\mathbf{t}_j}{\|\mathbf{t}_j\|_2}.$$

The cross-modal similarity is then

$$s_{ij} = \tau \, \hat{\mathbf{v}}_i^\top \hat{\mathbf{t}}_j, \tag{3}$$

where $\tau > 0$ is a learnable temperature. The same normalization is applied at evaluation (retrieval), keeping training and testing consistent.

**Soft-weighted contrastive loss.** With binary targets $y_{ij}$, which equals 1 if $i = j$ and 0 otherwise, the soft-weighted contrastive loss for one direction (e.g., V→T) is defined as:

$$\mathcal{L}_{\text{SWCA}}^{V \to T} = \frac{1}{B} \sum_{i=1}^{B} \sum_{j=1}^{B} (w_{ij}^{\text{Intra-sim}} + y_{ij}) \big[ -y_{ij} \log \sigma(s_{ij})$$
$$-(1 - y_{ij}) \log(1 - \sigma(s_{ij})) \big], \tag{4}$$

where $\sigma(\cdot)$ denotes the logistic sigmoid. The additive $y_{ij}$ term guarantees that exact matches are always prioritized, while the soft weights $w_{ij}^{\text{Intra-sim}}$ enable continuous supervision from partially similar pairs. To enforce mutual alignment, the loss is computed in both directions (V→T and T→V) and averaged:

$$\mathcal{L}_{\text{SWCA}}^{\text{bidirectional}} = \frac{1}{2} \left( \mathcal{L}_{\text{SWCA}}^{V \to T} + \mathcal{L}_{\text{SWCA}}^{T \to V} \right). \tag{5}$$

The similarity weight $w_{ij}^{\text{Intra-sim}}$ is computed twice under two complementary conditions: (i) a spatial coherence component $w_{ij}^{\text{spatial}}$ in spatially-guided semantic alignment, and (ii) a medical knowledge component $w_{ij}^{\text{knowledge}}$ in knowledge-infused semantic alignment.

### 3.3 SPATIALLY COHERENT SEMANTIC ALIGNMENT

Radiologists interpret CT scans as coherent 3D volumes rather than unordered collections of axial slices; spatial locality carries semantic importance. To encode this intuition into the alignment objective, we extend the soft-weighted contrastive loss with a *spatial similarity weight* $w_{ij}^{\text{spatial}}$ that emphasizes alignment consistent with 3D geometry.

Each volume $i$ is divided into $N$ cubic patches, with patch $m$ represented by its normalized centroid $\mathbf{c}_{i,m} \in [0,1]^3$. We compute a non-negative saliency score $r_{i,m}$ for each patch (e.g., based on feature norms or attention weights), and normalize:

$$\alpha_{i,m} \;=\; \frac{r_{i,m}}{\sum_{n=1}^{N} r_{i,n}}. \tag{6}$$

Using these coefficients, we summarize each scan with a weighted centroid ($\mu_i$) and a covariance descriptor:

$$\boldsymbol{\mu}_i \;=\; \sum_{m=1}^{N} \alpha_{i,m}\,\mathbf{c}_{i,m}, \qquad \boldsymbol{\Sigma}_i \;=\; \sum_{m=1}^{N} \alpha_{i,m}\,(\mathbf{c}_{i,m} - \boldsymbol{\mu}_i)(\mathbf{c}_{i,m} - \boldsymbol{\mu}_i)^{\top}, \tag{7}$$

where $\boldsymbol{\Sigma}_i$ is a $3 \times 3$ covariance matrix that captures the spread of the patch centroids. The spatial proximity between two volumes ($p_{ij}$) is then defined as:

$$p_{ij} \;=\; \exp\!\Big(-\frac{\|\boldsymbol{\mu}_i - \boldsymbol{\mu}_j\|_2^2}{2\kappa_\mu^2}\Big) \cdot \exp\!\Big(-\frac{\|\boldsymbol{\Sigma}_i - \boldsymbol{\Sigma}_j\|_F^2}{2\kappa_\Sigma^2}\Big), \tag{8}$$

where $\kappa_\mu$ and $\kappa_\Sigma$ control the decay with respect to centroid displacement and structural difference. These values are set based on physical CT spacing and embedding scale, specifically using the empirical standard deviations of the corresponding distance measures on a subset of the training data, preventing either term from dominating.

**Spatial similarity weight.** We combine this kernel with the intra-modal similarity weights $w_{ij}^{\text{Intra-sim}}$ (Equation (2)) via elementwise multiplication:

$$\delta_{ij} \;=\; w_{ij}^{\text{Intra-sim}}\, p_{ij}, \tag{9}$$

and row-normalize:

$$w_{ij}^{\text{spatial}} \;=\; \frac{\delta_{ij}}{\sum_k \delta_{ik} + \varepsilon}. \tag{10}$$

Replacing $w_{ij}^{\text{Intra-sim}}$ with $w_{ij}^{\text{spatial}}$ in Equation (4) and Equation (5) yields a spatially aware loss $\mathcal{L}_{\text{SWCA}}^{\text{spatial}}$. This formulation leverages centroid displacement and structural distribution to emphasize spatially coherent alignments.

### 3.4 KNOWLEDGE-INFUSED SEMANTIC ALIGNMENT

We incorporate medical knowledge into the learning process without increasing the number of trainable parameters by leveraging a frozen pre-trained medical language model, such as *HuatuoGPT-o1 7B* Chen et al. (2024a). This model encodes clinical reasoning, and understanding of medical terminology derived from large-scale medical corpora. Our framework remains model-agnostic and can support any comparable medical LLM such as LLaVA-Med Li et al. (2023a), Meditron Chen et al. (2023), or BioMistral Labrak et al. (2024b) with minimal changes.

**Medical-knowledge similarity weight.** Each medical report is passed through the selected medical LLM (e.g., HuatuoGPT) in inference mode. The final hidden states are mean-pooled and projected, yielding a fixed *knowledge embedding* $\mathbf{h}_i \in \mathbb{R}^d$ for sample $i$. We reuse the similarity formulation in Equation (1) and the soft normalization from Equation (2), where $\mathbf{z}$ now corresponds to the knowledge embeddings $\mathbf{h}$. This produces the semantic similarity weights $w_{ij}^{\text{knowledge}}$, which reflect report-level alignment. Substituting these into Equation (4) and Equation (5) defines the medical knowledge-infused loss $\mathcal{L}_{\text{SWCA}}^{\text{knowledge}}$, encouraging alignment across semantically related cases. It is noted that it goes beyond plain text-similarity reweighting: ontology features reshape the cross-modal target kernel, so CT embeddings are trained to reflect concept-level clinical structure.

### 3.5 OPTIMIZATION OBJECTIVE

The spatially-weighted and knowledge-weighted losses are merged with a convex combination:

$$\mathcal{L}_{\text{SWCA}}^{\text{spatial,knowledge}} = \alpha \, \mathcal{L}_{\text{SWCA}}^{\text{spatial}} + (1 - \alpha) \, \mathcal{L}_{\text{SWCA}}^{\text{knowledge}}, \alpha \in [0, 1], \tag{11}$$

where $\alpha$ balances pure spatially-guided semantic alignment against knowledge-infused semantic alignment. This unified objective integrates these two complementary semantics and continuous similarity into a single differentiable framework, producing representations that respect *where* abnormalities are located and *how* clinicians describe them.

## 4 EXPERIMENTS

We assess model performance across various evaluation scenarios using multiple objective metrics. Additional implementation details are provided in Section A.2.

### 4.1 DATASETS

We use two publicly available CT-report datasets: **CT-RATE** Hamamci et al. (2024a), which provides 50,188 reconstructed non-contrast chest CT volumes from 21,304 patients. We retain one reconstruction per scan, yielding 24,128 volumes for training, while the testing set of CT-RATE itself (1,564 volumes) is used as the in-domain test set. **BIMCV-R** Chen et al. (2024b), which contains 8,069 chest CT volumes paired with corresponding radiology reports, is used only for out-of-domain, zero-shot evaluation. For both datasets, scans are resampled to $256 \times 256 \times 32$ voxels, quantized to 8-bit integers, and stored in compressed NIfTI format.

### 4.2 MULTI-TASK ADAPTATION EVALUATION

As shown in Figure 4 in the appendix, for comprehensive benchmarking of SCALE-VLP, we conduct extensive evaluations on three downstream clinical applications: (1) CT-report cross-modal retrieval, (2) report generation, and (3) CT abnormality classification.

#### 4.2.1 CT-REPORT CROSS-MODAL RETRIEVAL

We evaluate retrieval in both *CT-to-report* and *report-to-CT* directions by ranking $\ell_2$-normalized embeddings using cosine similarity. All components of SCALE-VLP, including the 3D vision encoder, text encoder, and projection heads, are kept frozen during evaluation, as illustrated in Figure 4 in the appendix. Performance is reported as Recall@K, defined as the fraction of queries for which the correct match appears in the top-$K$ results. In addition, we report *SumR* (the sum of all available recalls per direction), which provides a single, scale-sensitive summary across cutoffs (Table 1).

As shown in Table 1, SCALE-VLP outperforms all state-of-the-art (SOTA) methods at every recall threshold and in both directions. At $N{=}100$, SCALE-VLP yields 10.0 points (IR) and 8.0 points (TR) improvements in R@1 compared to the strongest SOTA methods, respectively, and this advantage persists as the pool size grows. Even at $N{=}1564$, SCALE-VLP sustains its lead with the best performance, underscoring robustness at scale. While fVLM benefits from explicit anatomical priors through organ segmentation in pre-training, SCALE-VLP surpasses it, suggesting that our spatial– and knowledge–aware alignment effectively capture fine-grained clinical semantics.

#### 4.2.2 REPORT GENERATION

We begin with a pretrained 3D ViT encoder and BERT text encoder optimized via our SWCA objective. These frozen components encode CT–report pairs into semantically grounded embeddings with spatial structure and clinical semantics. All subsequent stages build on this frozen backbone:

- **Projection adaptation.** We develop a lightweight projection module $P_\theta : \mathbb{R}^{d_{\text{vis}}} \to \mathbb{R}^{d_{\text{llm}}}$ to map vision tokens into the latent space of a LLM (LLaMA-2-7B). Only $P_\theta$ and its input embeddings are updated, enabling volumetric grounding without disrupting the LLM's prior.

| N | Model | IR (CT → Report) | | | | | | TR (Report → CT) | | | | | |
|---|---|---|---|---|---|---|---|---|---|---|---|---|---|
| | | R@1 | R@5 | R@10 | R@50 | R@100 | SumR | R@1 | R@5 | R@10 | R@50 | R@100 | SumR |
| 100 | CT-CLIP Hamamci et al. (2024a) | 2.0 | 9.0 | 15.0 | 68.0 | — | 94.0 | 3.0 | 9.0 | 17.0 | 67.0 | — | 96.0 |
| | M3D Bai et al. (2024) | 3.0 | 9.0 | 16.0 | 69.0 | — | 97.0 | 3.0 | 8.0 | 20.0 | 74.0 | — | 105.0 |
| | SigLIP Zhai et al. (2023) | 1.0 | 6.0 | 12.0 | 55.0 | — | 74.0 | 1.0 | 5.0 | 10.0 | 52.0 | — | 68.0 |
| | Merlin Blankemeier et al. (2024) | 1.0 | 15.0 | 28.0 | 86.0 | — | 130.0 | 4.0 | 15.0 | 28.0 | 84.0 | — | 131.0 |
| | fVLM Shui et al. (2025) | 2.0 | 9.0 | 16.0 | 61.0 | — | 88.0 | 6.0 | 17.0 | 25.0 | 64.0 | — | 112.0 |
| | CWCL-Style | 6.0 | 27.0 | 40.0 | 79.0 | — | 152.0 | 9.0 | 29.0 | 44.0 | 80.0 | — | 162.0 |
| | SCALE-VLP w/o Spatial & Knowledge | 8.0 | 30.0 | 47.0 | 89.0 | — | 174.0 | 11.0 | 31.0 | 46.0 | 89.0 | — | 177.0 |
| | SCALE-VLP w/o Spatial | 12.0 | 37.0 | 55.0 | 93.0 | — | 197.0 | 13.0 | 40.0 | 54.0 | 92.0 | — | 199.0 |
| | SCALE-VLP w/o Knowledge | **17.0** | 36.0 | **56.0** | **96.0** | — | **205.0** | 10.0 | 37.0 | 58.0 | **94.0** | — | 199.0 |
| | **SCALE-VLP** | 13.0 | **40.0** | 56.0 | 94.0 | — | 203.0 | **14.0** | **42.0** | **59.0** | 93.0 | — | **208.0** |
| 500 | CT-CLIP Hamamci et al. (2024a) | 0.6 | 1.6 | 2.8 | 14.8 | 29.6 | 49.4 | 0.6 | 1.8 | 3.2 | 14.8 | 28.8 | 49.2 |
| | M3D Bai et al. (2024) | 0.8 | 1.6 | 3.2 | 15.2 | 28.8 | 49.6 | 1.0 | 3.0 | 5.4 | 21.0 | 37.2 | 67.6 |
| | SigLIP Zhai et al. (2023) | 0.2 | 1.4 | 2.6 | 12.0 | 21.6 | 37.8 | 0.2 | 1.0 | 2.0 | 10.2 | 20.6 | 34.0 |
| | Merlin Blankemeier et al. (2024) | 1.8 | 6.6 | 9.4 | 34.2 | 55.8 | 107.8 | 0.6 | 6.4 | 10.2 | 35.8 | 57.2 | 110.2 |
| | fVLM Shui et al. (2025) | 0.0 | 2.2 | 3.0 | 14.4 | 27.0 | 46.6 | 1.4 | 3.2 | 7.2 | 25.4 | 36.4 | 73.6 |
| | CWCL-Style | 2.6 | 6.2 | 12.4 | 34.2 | 52.8 | 108.2 | 2.6 | 7.8 | 11.4 | 35.4 | 53.4 | 110.6 |
| | SCALE-VLP w/o Spatial & Knowledge | 3.0 | 8.2 | 14.2 | 46.6 | 66.8 | 138.8 | 2.8 | 9.2 | 15.2 | 45.4 | 64.4 | 137.0 |
| | SCALE-VLP w/o Spatial | 3.2 | 11.6 | 17.8 | 49.4 | 69.4 | 151.4 | 3.2 | 11.2 | 18.8 | 50.4 | 69.0 | 152.6 |
| | SCALE-VLP w/o Knowledge | 2.2 | 11.2 | **20.2** | 52.2 | 71.6 | 157.4 | 3.2 | 10.4 | 18.0 | 51.2 | 71.4 | 154.2 |
| | **SCALE-VLP** | 3.4 | 12.2 | 19.8 | 53.2 | 72.6 | 161.2 | 4.4 | 11.8 | 21.0 | 51.6 | 72.6 | 161.4 |
| 1000 | CT-CLIP Hamamci et al. (2024a) | 0.3 | 1.0 | 2.0 | 7.8 | 15.1 | 26.2 | 0.4 | 1.0 | 1.8 | 7.8 | 14.9 | 25.9 |
| | M3D Bai et al. (2024) | 0.2 | 0.8 | 1.4 | 8.2 | 15.1 | 25.7 | 0.4 | 1.7 | 2.7 | 10.6 | 19.4 | 34.8 |
| | SigLIP Zhai et al. (2023) | 0.1 | 0.6 | 1.1 | 6.0 | 11.0 | 18.8 | 0.1 | 0.5 | 1.1 | 5.1 | 10.4 | 17.2 |
| | Merlin Blankemeier et al. (2024) | 1.2 | 3.1 | 5.1 | 19.1 | 33.6 | 62.1 | 0.9 | 2.7 | 4.5 | 20.0 | 34.5 | 62.6 |
| | fVLM Shui et al. (2025) | 0.2 | 0.9 | 1.4 | 7.1 | 13.5 | 23.1 | 0.5 | 2.1 | 3.4 | 13.9 | 22.0 | 41.9 |
| | CWCL-Style | 1.1 | 4.1 | 6.9 | 23.0 | 35.0 | 70.1 | 1.1 | 4.7 | 7.4 | 23.1 | 35.4 | 71.7 |
| | SCALE-VLP w/o Spatial & Knowledge | 1.5 | 5.0 | 8.5 | 28.3 | 46.0 | 89.3 | 1.3 | 5.5 | 9.2 | 26.9 | 44.6 | 87.5 |
| | SCALE-VLP w/o Spatial | 1.7 | 6.3 | 10.5 | 34.4 | 50.6 | 103.5 | 1.7 | 6.2 | 11.0 | 34.5 | 50.0 | 103.4 |
| | SCALE-VLP w/o Knowledge | 1.1 | 5.3 | 11.0 | 35.2 | **52.1** | 104.7 | 1.8 | 5.8 | 10.0 | 34.1 | **51.8** | 103.5 |
| | **SCALE-VLP** | 1.8 | 6.4 | 11.6 | 35.8 | 51.9 | 107.5 | 2.2 | 6.5 | 11.5 | 35.8 | 51.2 | 107.2 |
| 1564 | CT-CLIP Hamamci et al. (2024a) | 0.2 | 0.6 | 1.3 | 5.7 | 9.9 | 17.7 | 0.2 | 0.6 | 1.2 | 5.0 | 9.5 | 16.5 |
| | M3D Bai et al. (2024) | 0.1 | 0.6 | 0.9 | 4.7 | 10.1 | 16.4 | 0.3 | 1.0 | 1.7 | 7.0 | 12.9 | 22.9 |
| | SigLIP Zhai et al. (2023) | 0.1 | 0.4 | 0.7 | 3.8 | 7.4 | 12.4 | 0.1 | 0.3 | 0.6 | 3.3 | 6.5 | 10.8 |
| | Merlin Blankemeier et al. (2024) | 0.6 | 1.7 | 2.7 | 12.2 | 23.0 | 40.2 | 0.3 | 1.3 | 2.9 | 13.4 | 24.6 | 42.5 |
| | fVLM Shui et al. (2025) | 0.1 | 0.6 | 0.8 | 4.7 | 9.3 | 15.5 | 0.5 | 1.8 | 2.8 | 9.9 | 16.6 | 31.6 |
| | CWCL-Style | 0.6 | 3.1 | 4.5 | 16.1 | 25.5 | 49.8 | 0.6 | 3.0 | 5.0 | 17.1 | 25.6 | 51.3 |
| | SCALE-VLP w/o Spatial & Knowledge | 1.0 | 3.7 | 6.1 | 20.5 | 33.1 | 64.4 | 1.0 | 3.6 | 6.0 | 20.4 | 32.0 | 63.0 |
| | SCALE-VLP w/o Spatial | 1.0 | **4.2** | 7.1 | 24.9 | 38.2 | 75.4 | 1.0 | 4.0 | 7.1 | 24.4 | 38.1 | 74.6 |
| | SCALE-VLP w/o Knowledge | 1.1 | 3.1 | 7.0 | **25.2** | **39.7** | 76.1 | 1.2 | 3.6 | 6.5 | 25.0 | **39.2** | 75.5 |
| | **SCALE-VLP** | 1.2 | 4.2 | 7.4 | 25.1 | 38.8 | 76.7 | 1.3 | 4.2 | 7.4 | 25.4 | 38.5 | 76.8 |

Table 1: **CT-report cross-modal retrieval results on CT-RATE.** Illustration of results across recall levels, retrieval directions, and pool sizes $N \in \{100, 500, 1000, 1564\}$ for all models. The pool size N is the number of candidate CT-report pairs evaluated during retrieval (1 correct pair + N-1 samples)

- **Low-rank adaptation.** To generate full radiology reports, we insert rank-16 LoRA adapters (Hu et al., 2022) into each decoder layer and fine-tune the model using cross-entropy loss over report sequences. The vision encoder and projector remain frozen throughout, and only the LoRA and projection parameters are updated. By building on pretrained alignment and restricting updates to a minimal set of components, our approach enables efficient report generation with strong performance and low memory overhead.

Table 2 shows that SCALE-VLP achieves consistent improvements over SOTA methods across all nine metrics. The most notable gain is on BLEU-4, which nearly doubles compared to CT-CLIP (∼2× improvement). Comparable relative gains are observed on semantically oriented metrics: ROUGE-L improves by 56%, METEOR by 50%, and CIDEr–D more than doubles. Against the stronger M3D baseline, SCALE-VLP still yields sizable advantages, including +82% on BLEU-4, +42% on ROUGE-L, and +45% on METEOR, with BERT-F1 also increasing to 0.8934. Together, demonstrating that SCALE-VLP 's embedding captures rich semantic information.

Table 2 shows that SCALE-VLP achieves consistent improvements over SOTA methods across all ten metrics. The most notable gain is on BLEU-4, which nearly doubles compared to CT-CLIP (∼2× improvement). Comparable relative gains are observed on semantically oriented metrics: ROUGE-L improves by 56%, METEOR by 50%, and CIDEr–D more than doubles. Against the stronger M3D baseline, SCALE-VLP still yields sizable advantages, including +82% on BLEU-4, +42% on ROUGE-L, +45% on METEOR, and GREEN increasing to 0.4054, with BERT-F1 also increasing to 0.8934. Together, demonstrating that SCALE-VLP 's embedding captures rich semantic information.

| Model | BLEU-1 | BLEU-2 | BLEU-3 | BLEU-4 | ROUGE-1 | ROUGE-L | METEOR | BERT-F1 | CIDEr-D | GREEN |
|---|---|---|---|---|---|---|---|---|---|---|
| CT-CLIP Hamamci et al. (2024a) | 0.3681 | 0.2759 | 0.2179 | 0.1766 | 0.4257 | 0.2823 | 0.3140 | 0.8582 | 0.0837 | 0.3721 |
| M3D Bai et al. (2024) | 0.3468 | 0.2620 | 0.2067 | 0.1695 | 0.4792 | 0.3107 | 0.3246 | 0.8711 | 0.1181 | 0.3955 |
| Merlin Blankemeier et al. (2024) | 0.4428 | 0.1864 | 0.1035 | 0.0548 | 0.1112 | 0.0854 | 0.0221 | 0.8211 | 0.0104 | 0.3805 |
| SCALE-VLP w/o Spatial & Knowledge | 0.4124 | 0.2572 | 0.2135 | 0.1833 | 0.4894 | 0.3237 | 0.3791 | 0.8696 | 0.1052 | – |
| SCALE-VLP w/o Spatial | 0.4367 | 0.3479 | 0.2882 | 0.2454 | 0.5303 | 0.3690 | 0.3985 | 0.8784 | 0.1273 | – |
| **SCALE-VLP** | **0.5210** | **0.4433** | **0.3886** | **0.3485** | **0.5984** | **0.4408** | **0.4709** | **0.8934** | **0.1684** | **0.4054** |

Table 2: **Report generation results on CT-RATE.**

| Model | (A) Summary Metrics | | | (B) Cluster-average Accuracy | | | | | |
|---|---|---|---|---|---|---|---|---|---|
| | Accuracy | F1 | AUC | Airway / Bronchi | Alveolar / Airspace | Interstitial | Pleural / Extra-pulmonary | Nodular / Mass | Vascular / Cardiac |
| CT-CLIP Hamamci et al. (2024a) | 0.41 | 0.53 | **0.52** | 0.31 | 0.46 | 0.29 | 0.38 | 0.38 | 0.51 |
| M3D Bai et al. (2024) | 0.48 | 0.58 | 0.48 | **0.89** | 0.43 | 0.29 | 0.11 | 0.61 | 0.47 |
| Merlin Blankemeier et al. (2024) | 0.62 | 0.50 | 0.50 | 0.80 | 0.66 | 0.64 | 0.36 | 0.54 | 0.67 |
| fVLM Shui et al. (2025) | 0.69 | 0.58 | 0.51 | **0.89** | **0.77** | **0.71** | 0.46 | 0.43 | **0.79** |
| SCALE-VLP w/o Spatial & Knowledge | 0.49 | 0.59 | 0.50 | 0.45 | 0.62 | 0.30 | 0.69 | 0.50 | 0.31 |
| SCALE-VLP w/o Spatial | 0.53 | 0.59 | 0.50 | 0.77 | 0.47 | 0.53 | 0.50 | 0. 43 | 0.50 |
| **SCALE-VLP** | **0.72** | **0.59** | **0.52** | 0.87 | 0.71 | 0.51 | **0.76** | **0.56** | 0.78 |

Table 3: **CT abnormality classification results on CT-RATE.**

### 4.2.3 CT ABNORMALITY CLASSIFICATION

We evaluate the transferability of the frozen image encoder to supervised multi-label classification across 13 thoracic findings from CT-RATE. A Multi-Task classification head is trained on top of projector. To support interpretability, we also group findings into six clinical clusters (airway, alveolar, interstitial, pleural, nodular, vascular) based on anatomical and pathophysiological coherence. Global and cluster-level results are shown in Table 3, with full per-task scores in Appendix Table 6. Despite fVLM being trained with organ segmentations and therefore benefiting from explicit biological priors, SCALE-VLP achieves stronger overall accuracy (0.72), F1 (0.59) and AUC (0.52), outperforming fVLM (0.69 / 0.58 / 0.50). At the cluster level, M3D peaks on airway (0.89), fVLM leads interstitial (0.71) and alveolar (0.77), while SCALE-VLP is the most balanced, surpassing 0.70 in four clusters.

### 4.3 CROSS-DOMAIN GENERALIZATION EVALUATION

We evaluate zero-shot generalization on the external BimCV-R dataset under domain shift with all parameters frozen. As shown in Table 4, SCALE-VLP consistently matches or outperforms SOTA methods across both retrieval and generation tasks. In cross-modal retrieval, it ties the best performance at small $K$ while achieving clear gains at higher cutoffs, leading to the highest overall SumR across both pool sizes (e.g., +6 points over Merlin at CT $\rightarrow$ Report, $N=100$ ). For report generation, SCALE-VLP achieves the best results across BLEU, ROUGE, METEOR, and BERT-F1. Together, the consistent improvements in both retrieval and report generation tasks highlight the robustness of the learned volumetric and semantic representations, enabling effective zero-shot transfer to unseen scanners and populations.

| | Retrieval (N = 100) | | | | | | | | Retrieval (N = 1000) | | | | | | | | | Report Generation | | | |
|---|---|---|---|---|---|---|---|---|---|---|---|---|---|---|---|---|---|---|---|---|---|
| | CT $\rightarrow$ Report | | | | Report $\rightarrow$ CT | | | | CT $\rightarrow$ Report | | | | | Report $\rightarrow$ CT | | | | | BLEU | ROUGE | METEOR | BERT-F1 |
| Model | R@1 | R@5 | R@10 | sumR | R@1 | R@5 | R@10 | sumR | R@1 | R@5 | R@10 | R@50 | R@100 | SumR | R@5 | R@10 | R@50 | R@100 | SumR | | | | |
| CT-CLIP (Hamamci et al., 2024a) | 1.0 | 6.0 | 11.0 | 18.0 | 1.0 | 5.0 | 10.0 | 16.0 | 0.6 | 1.3 | 6.0 | 11.0 | 18.9 | 0.5 | 1.1 | 6.0 | 11.0 | 18.6 | 0.2022 | 0.1939 | 0.1047 | 0.8082 |
| M3D (Bai et al., 2024) | 2.0 | 6.0 | 10.0 | 18.0 | 1.0 | 5.0 | 12.0 | 18.0 | 0.7 | 1.4 | 6.3 | 11.5 | 19.9 | 0.6 | 1.6 | 6.5 | 12.2 | 20.9 | 0.2044 | 0.1965 | 0.1078 | 0.8106 |
| Merlin (Blankemeier et al., 2024) | **4.0** | 8.0 | 11.0 | 23.0 | **2.0** | 5.0 | 13.0 | 20.0 | **0.9** | 1.6 | 7.0 | 13.0 | 22.5 | **1.2** | 1.7 | 7.2 | 13.0 | 23.1 | 0.2188 | 0.1309 | 0.0310 | 0.8125 |
| fVLM (Shui et al., 2025) | 1.0 | 5.0 | 10.0 | 16.0 | **2.0** | 5.0 | 13.0 | 20.0 | 0.8 | 1.2 | 5.2 | 9.7 | 16.9 | 0.7 | 1.4 | 4.9 | 10.0 | 17.0 | — | — | — | — |
| **SCALE-VLP** | **4.0** | **10.0** | **15.0** | **29.0** | **2.0** | **7.0** | **14.0** | **23.0** | 0.8 | **2.1** | **7.7** | **13.3** | **23.9** | **1.2** | **2.0** | **7.5** | **13.9** | **24.6** | **0.2406** | **0.2231** | **0.1416** | **0.8220** |

Table 4: **Zero-shot results on BimCV-R**: retrieval (N=100, N=1000) and report generation.

### 4.4 ABLATION STUDY

We conduct ablation studies to evaluate the contribution of individual components in our framework, the effect of them at varying training data ratios, and the value of the mixing parameter $\alpha$.

#### 4.4.1 SPATIAL AND KNOWLEDGE ALIGNMENT

We study the contributions of spatial and knowledge cues through ablations. As shown in Table 1, removing spatial alignment (i.e., SWCA-base + Knowledge) reduces *SumR* by an average 5.6%

| N | Model | IR (CT → Report) | | | | | | TR (Report → CT) | | | | | |
|---|---|---|---|---|---|---|---|---|---|---|---|---|---|
| | | R@1 | R@5 | R@10 | R@50 | R@100 | SumR | R@1 | R@5 | R@10 | R@50 | R@100 | SumR |
| 100 | SCALE-VLP (HuatuoGPT) | 13.0 | 40.0 | 56.0 | 94.0 | — | 203.0 | 14.0 | 42.0 | 59.0 | 93.0 | — | 208.0 |
| | SCALE-VLP (LLaMA3-Med42) | 11.0 | 37.0 | 55.0 | 97.0 | — | 200.0 | 11.0 | 34.0 | 58.0 | 94.0 | — | 197.0 |
| | SCALE-VLP (BioMistral) | 15.0 | 37.0 | 53.0 | 97.0 | — | 202.0 | 15.0 | 35.0 | 55.0 | 96.0 | — | 201.0 |
| 500 | SCALE-VLP (HuatuoGPT) | 3.4 | 12.2 | 19.8 | 53.2 | 72.6 | 161.2 | 4.4 | 11.8 | 21.0 | 51.6 | 72.6 | 161.4 |
| | SCALE-VLP (LLaMA3-Med42) | 3.6 | 10.6 | 18.6 | 55.0 | 73.0 | 160.8 | 3.0 | 11.0 | 19.2 | 53.4 | 73.4 | 160.0 |
| | SCALE-VLP (BioMistral) | 3.4 | 11.0 | 18.2 | 53.4 | 71.4 | 157.4 | 4.0 | 13.4 | 20.2 | 52.6 | 71.2 | 161.4 |
| 1000 | SCALE-VLP (HuatuoGPT) | 1.8 | 6.4 | 11.6 | 35.8 | 51.9 | 107.5 | 2.2 | 6.5 | 11.5 | 35.8 | 51.2 | 107.2 |
| | SCALE-VLP (LLaMA3-Med42) | 1.8 | 5.6 | 10.1 | 35.0 | 52.5 | 105.0 | 1.5 | 5.6 | 11.1 | 34.2 | 51.4 | 103.8 |
| | SCALE-VLP (BioMistral) | 1.6 | 5.6 | 10.0 | 34.1 | 52.5 | 103.8 | 1.9 | 6.8 | 10.7 | 34.8 | 51.6 | 105.8 |
| 1564 | SCALE-VLP (HuatuoGPT) | 1.2 | 4.2 | 7.4 | 25.1 | 38.8 | 76.7 | 1.3 | 4.2 | 7.4 | 25.4 | 38.5 | 76.8 |
| | SCALE-VLP (LLaMA3-Med42) | 1.2 | 4.3 | 7.2 | 24.6 | 39.4 | 76.7 | 1.1 | 4.0 | 7.5 | 24.4 | 38.5 | 75.4 |
| | SCALE-VLP (BioMistral) | 0.9 | 3.8 | 7.0 | 23.8 | 39.1 | 74.5 | 1.2 | 3.9 | 6.8 | 24.7 | 40.0 | 76.6 |

Table 5: **Sensitivity of SCALE-VLP to the choice of medical VLM.** Retrieval performance on CT-RATE for different medical backbones used in the knowledge-infused term.

drop across pool sizes (IR and TR), while removing knowledge cues (i.e., SWCA-base + Spatial) demonstrates a lower average (3.3%) drop compared to the full SWCA version. Finally, SWCA without both spatial and knowledge alignment (i.e., SWCA-base) yields the largest drop (21.3%). Both spatial reasoning and medical knowledge provide substantial, complementary contributions to cross-modal alignment, with their combination consistently yielding the best performance. For report generation (Table 2), removing spatial alignment lowers BLEU-4 by ∼30% and METEOR by ∼15%, while dropping both spatial and knowledge cues reduces BLEU-4 by ∼47% and METEOR by ∼20%. These degradations highlight that both spatial coherence and knowledge priors are important, and that combining them leads to the largest gains in fluency and fidelity of the generated reports.

For classification task, ablations in Table 3 show that removing spatial alignment or both spatial and knowledge alignment consistently reduces performance across clinical clusters. These results confirm that both components are essential for robust cross-task transfer, with spatial alignment as the dominant factor and knowledge cues providing complementary benefits. By jointly modeling both, SCALE-VLP achieves SOTA performance across retrieval, report generation, and classification, establishing a unified framework for CT-RATE.

### 4.4.2 Role of Different External Medical VLMs for Knowledge Fusion

We expand our evaluation to include additional external VLM models to evaluate the sensitivity of SCALE-VLP to the utilized medical knowledge VLM. To that end, we conduct a study evaluating three different medical VLMs: HuatuoGPT (default), LLaMA3-Med42-8B Christophe et al. (2024), and BioMistral-7B Labrak et al. (2024a). The results are available in Table 5. The results demonstrate that the proposed framework generalizes to different Medical VLMs, and retrieval performance remains highly consistent across all three models for all pool sizes. Additionally, while there are only marginal variations between them, all variants substantially outperform the evaluated baselines. This consistency indicates that SCALE-VLP is not sensitive to any single VMM's internal knowledge distribution, bias, or reasoning patterns. Instead, the utilized VLM functions as a modular knowledge prior, whose role is to provide coarse semantic guidance rather than to fully dominate the alignment process.

### 4.4.3 Role of SWCA in Large-Batch Scaling and Consistent Gains

Our SWCA replaces the softmax-based InfoNCE with a pairwise sigmoid formulation, which eliminates the need to materialize the full $B \times B$ similarity matrix and substantially reduces memory overhead. This design keeps VRAM usage within budget and shortens alignment time, enabling the *base model* to scale the per-device and effective batch size. Figure 2 compares the *base model* against our SWCA-only variant, *SCALE-VLP w/o Spatial & Knowledge alignment*. The results highlight the effect of SWCA, where training with larger batches and soft-weighted similarities yields consistent improvements across all sampling fractions relative to the *base model*. Furthermore, Figure 2 illustrates retrieval performance as a function of training samples.

While *base model* retrieval performance stagnates throughout training, our SWCA variants continue to improve as they are exposed to more data. Adding the knowledge-alignment head (*SCALE-VLP w/o Spatial alignment*) provides additional gains, and the full *SCALE-VLP*, with both knowledge and spatial heads, achieves the strongest performance, and shows the highest growth as training scales. In addition, by replacing binary supervision with continuous targets derived from CT–CT and report–report similarities, each sample contributes graded supervision to many CT-report pairs instead of only one. In medical applications with a scarcity of paired data compared to the general domain, this design allows information aggregation from multiple samples instead of only a single positive sample.

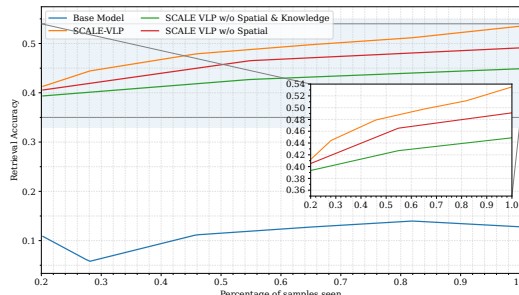

Figure 2: Batch scaling effect

### 4.4.4 MIXING PARAMETER $\alpha$.

Across Tables 7 to 10, we study the effect of the mixing parameter $\alpha$. Performance differences between $\alpha$ values (0.3–0.6) are modest but consistent, typically ranging from 0.2 to 1.5 points. Based on these results and the trends in Figures 5 and 6, we adopt $\alpha=0.5$ as the default for all subsequent experiments.

## 4.5 QUALITATIVE VISUALIZATION OF ALIGNMENT

We qualitatively assess alignment and clustering properties of SCALE-VLP on the CT-RATE validation set. Figure 3 summarizes both accuracy distribution across abnormality clusters and embedding similarity patterns. Panel (**a**) reports accuracy across clinically defined clusters including airway/bronchi, alveolar/airspace, interstitial, pleural, nodular, and vascular groups, which are also used to organize the similarity matrices in panels (**b**) and (**c**). Panels (**b**) and (**c**) visualize pairwise cosine similarity between CT and report embeddings. When sorted by unsupervised k-means clusters (**b**), block diagonal structure emerges, reflecting latent grouping without external supervision. When reordered

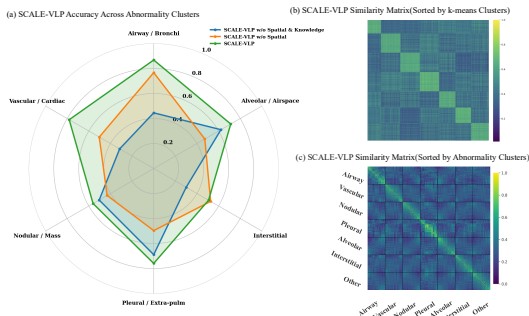

Figure 3: Similarity Matrices by Model and Cluster. Panels (b–c) share axes: x = report embeddings; y = CT scan embeddings.

by the predefined abnormality categories (**c**), coherent clusters corresponding to abnormality categories become evident. Together, these results demonstrate that SCALE-VLP learns embedding spaces that reflect pathology aware structures.

## 5 CONCLUSION

SCALE-VLP pioneers a unified vision-language framework for 3D medical imaging by integrating spatial coherence and medical knowledge through soft-weighted contrastive learning. It overcomes critical limitations in volumetric data analysis: i) Data scarcity via continuous affinity modeling, ii) Spatial fragmentation through geometry-aware kernels, and iii) Medical knowledge neglect via domain-specific knowledge embedding fusion. Validated on CT-RATE and BimCV-R benchmarks, SCALE-VLP achieves SOTA performance on CT-report cross-modal retrieval, report generation, and abnormality classification, demonstrating SCALE-VLP possesses strong robustness and generalization capabilities.

## REPRODUCIBILITY STATEMENT

Implementation details for SCALE-VLP appear in Section A.2; dataset curation and CT-RATE/BimCV-R preprocessing are in Section 4.1 and Section A.1, respectively. The code, checkpoints, and evaluation scripts will be released upon acceptance.

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

Figure 4: Framework of multi-task SCALE-VLP for clinical downstream tasks: (1) report generation, (2) CT–report retrieval, and (3) CT abnormality classification.

# A APPENDIX

## A.1 PRE-PROCESSING PIPELINE

Prior to training, all CT volumes undergo a deterministic and reproducible pre-processing pipeline:

- **Spatial normalization.** Each scan is resampled to a fixed spatial resolution of $256 \times 256 \times 32$ voxels using trilinear interpolation, preserving the native affine transformation and axial orientation.
- **Intensity quantization.** Voxel intensities are cast to `float16` and linearly quantized to signed 8-bit integers. This preserves clinically meaningful contrast while reducing memory consumption by approximately $4\times$.
- **Storage and efficiency.** Volumes are saved in compressed NIfTI format to prevent partial writes. Parallel processing with eight workers reduces the average scan size by $6\times$.

This pipeline is fully reproducible: repeated runs over the same dataset release produce identical output hashes, ensuring consistent and verifiable results.

## A.2 IMPLEMENTATION DETAILS

All experiments are conducted on a Slurm-managed cluster with four NVIDIA A40 GPUs (48 GB each), 14 CPU cores, and 160 GB RAM. We utilize PyTorch 2.3 with HuggingFace ACCELERATE in `bf16` mixed precision, and configure training using four processes. Experiment tracking and checkpoint management are handled via Weights & Biases. For contrastive pre-training, SCALE-VLP operates with 55 CT–report pairs per GPU, resulting in an effective batch size of 220. Data loading employs two worker threads and pinned memory. No gradient accumulation is used during training. We set the loss interpolation parameters to $\alpha = 0.5$ based on retrieval performance observed in Sec. Tables 7 to 10. We employ the AdamW optimizer with a weight decay of 0.1, cosine learning rate scheduling, 3% linear warm-up, and gradient clipping at 0.5. The learning rate is set to $1 \times 10^{-4}$ for pre-training and $5 \times 10^{-5}$ for fine-tuning. Fine-tuning is performed using Deep-Speed ZeRO-3 with a per-GPU batch size of 8, updating only task-specific heads while keeping the encoders frozen. Pre-training for 10 epochs takes 3.8 hours for SCALE-VLP, whereas CT-CLIP, M3D, and Merlin require 5.3, 5.58, and 10 hours, respectively. We have trained M3D, CT-CLIP, and SigLIP under identical settings to ensure comparability. For fVLM, we have used the officially released pre-trained weights on CT-RATE. Fine-tuning SCALE-VLP for report generation for 4 epochs requires an additional 3 hours on average. As the weights, code, and configurations of fVLM's report generation model are not publicly available, we restrict our evaluation of this model to retrieval and abnormality classification tasks.

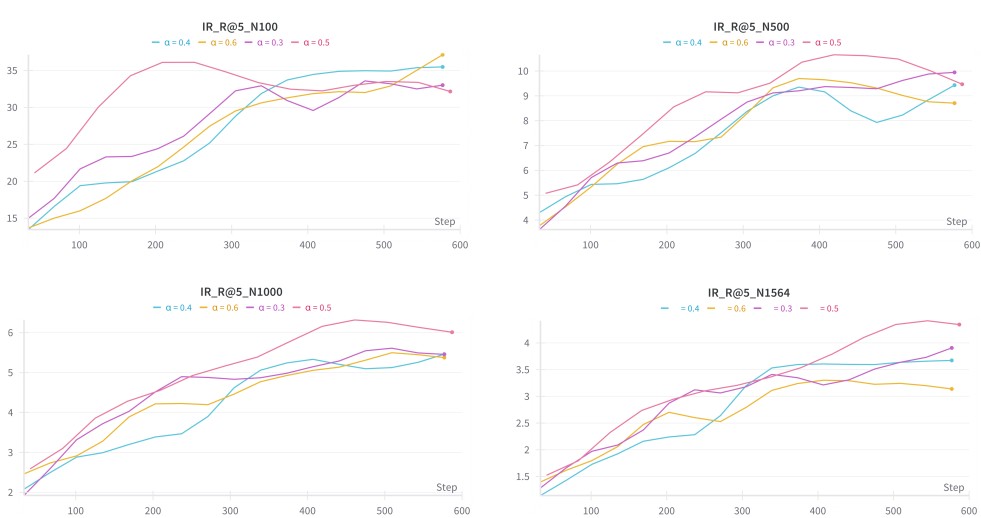

Figure 5: Effect of the mixing parameter $\alpha$ on CT $\rightarrow$ Report retrieval.

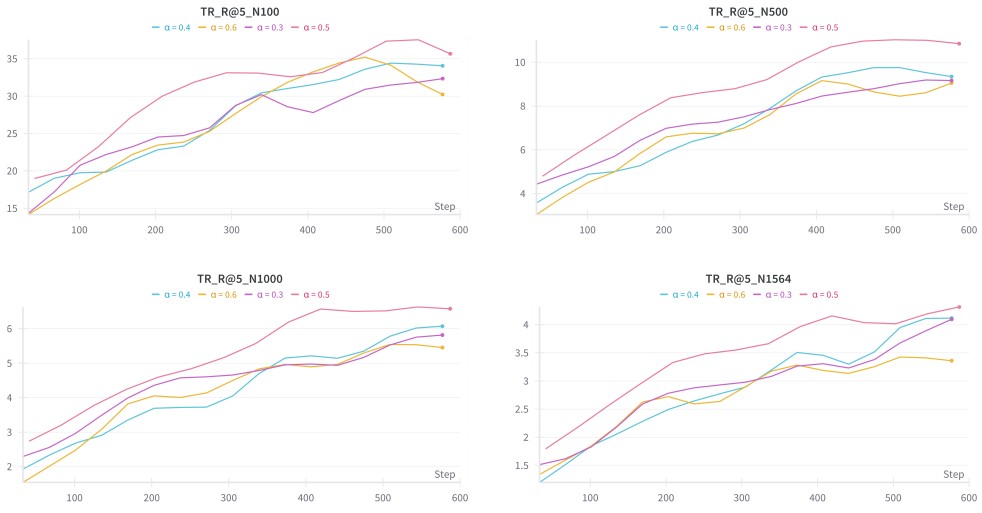

Figure 6: Effect of the mixing parameter $\alpha$ on Report $\rightarrow$ CT retrieval.

## A.3 PER-TASK ABNORMALITY CLASSIFICATION ON CT-RATE

Table 6 reports per-task abnormality classification accuracy on CT-RATE for SCALE-VLP and all baselines. CT-CLIP and M3D models show unbalanced behavior as CT-CLIP performs relatively well on calcification-related tasks (e.g., arterial wall, coronary artery) but struggles with effusion and consolidation, whereas M3D achieves strong results on cardiomegaly, peribronchial thickening, and bronchiectasis but fails on mosaic attenuation, pericardial effusion, hiatal hernia, and coronary artery wall calc. Among recent baselines, Merlin shows stronger performance on most tasks, such as cardiomegaly (0.90), peribronchial thickening (0.89), coronary artery wall calc (0.70), consolidation(79), but remains weaker on some tasks, such as pericardial effusion (0.08). fVLM is trained with additional organ segmentation data and therefore benefits from explicit biological priors, and

Table 6: Per-task accuracy on CT-RATE.

| Task | CT-CLIP | M3D | Merlin | fVLM | SCALE-VLP w/o Spatial & Knowledge | SCALE-VLP w/o Spatial | **SCALE-VLP** |
|---|---|---|---|---|---|---|---|
| Arterial wall calcification | 0.72 | 0.28 | 0.40 | 0.72 | 0.29 | 0.66 | 0.70 |
| Cardiomegaly | 0.45 | 0.90 | 0.90 | 0.90 | 0.42 | 0.10 | 0.90 |
| Pericardial effusion | 0.08 | 0.08 | 0.08 | 0.08 | 0.54 | 0.14 | 0.67 |
| Coronary artery wall calc. | 0.37 | 0.24 | 0.70 | 0.76 | 0.24 | 0.74 | 0.75 |
| Hiatal hernia | 0.68 | 0.15 | 0.65 | 0.85 | 0.84 | 0.85 | 0.85 |
| Lymphadenopathy | 0.25 | 0.75 | 0.59 | 0.38 | 0.52 | 0.34 | 0.62 |
| Lung nodule | 0.50 | 0.48 | 0.49 | 0.48 | 0.48 | 0.52 | 0.50 |
| Lung opacity | 0.45 | 0.42 | 0.52 | 0.58 | 0.42 | 0.42 | 0.42 |
| Pulmonary fibrotic sequela | 0.29 | 0.29 | 0.64 | 0.71 | 0.30 | 0.53 | 0.51 |
| Mosaic attenuation pattern | 0.73 | 0.07 | 0.66 | 0.93 | 0.79 | 0.79 | 0.93 |
| Peribronchial thickening | 0.33 | 0.89 | 0.89 | 0.89 | 0.62 | 0.64 | 0.89 |
| Consolidation | 0.21 | 0.79 | 0.79 | 0.79 | 0.66 | 0.21 | 0.79 |
| Bronchiectasis | 0.29 | 0.89 | 0.72 | 0.89 | 0.27 | 0.89 | 0.85 |

Table 7: Retrieval at pool size $N{=}100$ for different $\alpha$ values.

| $\alpha$ | **IR** (CT $\rightarrow$ Report) | | | | **TR** (Report $\rightarrow$ CT) | | | |
|---|---|---|---|---|---|---|---|---|
| | R@1 | R@5 | R@10 | R@50 | R@1 | R@5 | R@10 | R@50 |
| 0.3 | 13.0 | 38.0 | 55.0 | 93.0 | 11.0 | 36.0 | 56.0 | 94.0 |
| 0.4 | 12.0 | 39.0 | 55.0 | 95.0 | 14.0 | 40.0 | 56.0 | 93.0 |
| 0.5 | **15.0** | **40.0** | **58.0** | **97.0** | **15.0** | **41.0** | **58.0** | **96.0** |
| 0.6 | 12.0 | 37.0 | 56.0 | 95.0 | 12.0 | 38.0 | 56.0 | 94.0 |

Table 8: Retrieval at pool size $N{=}500$ for different $\alpha$ values.

| $\alpha$ | **IR** (CT $\rightarrow$ Report) | | | | | **TR** (Report $\rightarrow$ CT) | | | | |
|---|---|---|---|---|---|---|---|---|---|---|
| | R@1 | R@5 | R@10 | R@50 | R@100 | R@1 | R@5 | R@10 | R@50 | R@100 |
| 0.3 | 2.8 | 10.6 | 19.0 | 52.2 | 70.4 | 3.2 | 9.8 | 17.0 | 50.8 | 69.4 |
| 0.4 | 3.2 | 10.2 | 19.0 | 51.2 | 71.8 | 3.4 | 10.6 | 18.0 | 51.2 | **71.2** |
| 0.5 | **3.4** | **12.0** | 19.0 | **55.2** | **72.4** | **4.8** | **12.6** | **20.4** | **52.4** | **71.2** |
| 0.6 | 3.0 | 11.6 | 19.0 | 51.8 | 71.4 | 3.0 | 11.2 | 18.8 | 50.3 | 70.4 |

this improved fVLM's performance compared to other baselines. In contrast, SCALE-VLP, without using any auxiliary data such as segmentations, achieves consistently competitive or superior results across categories. It matches or exceeds the best prior methods on high-prevalence findings such as hiatal hernia (0.85), mosaic attenuation (0.93), and peribronchial thickening (0.89), while also offering improvements on more challenging cases such as pericardial effusion (0.67). These results combined with the overall performance shown in Table 3 indicate that integrating SWCA, and spatial and knowledge alignment balances the strengths of prior models and yields robust performance across diverse clinical abnormalities.

A.4 ABLATION ON THE MIXING PARAMETER $\alpha$

Across Tables 7 to 10, we examine how the mixing parameter $\alpha$ influences performance. Although differences among values in the 0.3–0.6 range are relatively small, they are consistent, typically spanning 0.2–1.5 points. Considering these results alongside the trends in Figures 5 and 6, we select $\alpha{=}0.5$ as a balanced choice and use it as the default in all subsequent experiments.

A.5 QUALITATIVE EVALUATION OF REPORT GENERATION

To qualitatively assess report generation, we compare outputs from SCALE-VLP, M3D, and CT-CLIP against the ground truth for a representative patient case (Figure Figure 7). All models correctly capture core thoracic findings such as bilateral pleural effusions, associated compressive or dependent atelectasis, interstitial or septal thickening, ground-glass opacities, and cardiomegaly.

Table 9: Retrieval at pool size $N$=1000 for different $\alpha$ values.

| $\alpha$ | **IR** (CT → Report) | | | | | **TR** (Report → CT) | | | | |
|---|---|---|---|---|---|---|---|---|---|---|
| | R@1 | R@5 | R@10 | R@50 | R@100 | R@1 | R@5 | R@10 | R@50 | R@100 |
| 0.3 | 1.9 | 6.0 | 11.2 | 34.1 | 51.3 | 1.6 | 6.3 | 10.3 | 34.0 | 50.0 |
| 0.4 | 1.6 | 5.8 | 10.7 | 35.0 | 51.9 | 1.8 | 6.6 | 11.0 | 34.3 | 50.1 |
| 0.5 | **2.1** | **6.9** | **11.6** | **35.7** | **53.1** | **1.9** | **7.4** | **11.6** | **35.1** | **51.2** |
| 0.6 | 1.5 | 6.0 | 11.2 | 34.7 | 51.4 | 1.4 | 6.6 | 10.7 | 33.9 | 50.6 |

Table 10: Retrieval at pool size $N$=1564 for different $\alpha$ values.

| $\alpha$ | **IR** (CT → Report) | | | | | **TR** (Report → CT) | | | | |
|---|---|---|---|---|---|---|---|---|---|---|
| | R@1 | R@5 | R@10 | R@50 | R@100 | R@1 | R@5 | R@10 | R@50 | R@100 |
| 0.3 | 1.2 | 4.3 | 7.4 | 24.4 | 37.9 | 1.1 | 4.3 | 7.2 | 24.1 | 37.2 |
| 0.4 | 1.1 | 4.2 | 7.6 | 25.3 | 39.7 | **1.2** | 4.5 | 7.4 | 24.7 | 38.0 |
| 0.5 | **1.5** | **4.6** | **7.7** | **26.3** | **40.0** | **1.2** | **4.7** | **8.1** | **25.8** | **38.8** |
| 0.6 | 0.9 | 3.6 | 6.8 | 25.0 | 38.9 | 0.9 | 3.9 | 6.3 | 23.5 | 38.0 |

However, SCALE-VLP most closely mirrors the ground-truth description, accurately localizing effusions, identifying the port catheter tip within the right atrium, and describing diffuse lung pathology with high fidelity. In contrast, M3D and CT-CLIP omit certain device-related and anatomical details, with CT-CLIP notably failing to mention the catheter. None of the models capture more subtle findings such as minimal pericardial effusion or aortic calcifications. While SCALE-VLP introduces a minor hallucinated finding (a renal calculus), it still achieves the highest alignment with the reference report in terms of content coverage and clinical relevance. This is reflected in its superior language metrics (BLEU-4: 0.373, ROUGE-1: 0.596, METEOR: 0.475), indicating stronger semantic and syntactic coherence.

### A.6 QUALITATIVE VISUALIZATION OF EMBEDDINGS

We qualitatively assess the t-SNE projections on the CT-RATE validation set. In Figure 8, we project each sample's volume embeddings via t-SNE. SCALE-VLP forms distinct, compact clusters (Figure 8.(**c**)), whereas CT-CLIP (Figure 8.(**a**)) produces coarse grouping and M3D (Figure 8.(**b**)) shows unstructured scatter. Considering the similarity matrix visualization at Figure 3 and t-SNE plot together, these visualizations demonstrate that SCALE-VLP achieves superior semantic alignment and clustering compared to prior models.

We qualitatively evaluate representation quality using t-SNE projections on the CT-RATE validation set. As shown in Figure 8, we embed each sample's volume representation via t-SNE. SCALE-VLP forms distinct and compact clusters (Figure 8.**c**), whereas CT-CLIP (Figure 8.**a**) yields coarse groupings and M3D (Figure 8.**b**) produces scattered, unstructured points. When viewed alongside the similarity matrices in Figure 3, these visualizations highlight that SCALE-VLP achieves stronger semantic alignment and more coherent clustering compared to prior models.

## B USE OF LARGE LANGUAGE MODELS

We used a Large Language Model (LLM) solely to aid in polishing the writing of this paper. Specifically, the LLM was employed to correct grammar, fix typographical errors, and improve the clarity of expression; the authors verified all content.

**Sample**

**Ground truth:**
Findings: Bilateral pleural effusion is observed, more prominent on the right side, associated with surrounding atelectasis. Interlobular septal thickening and diffuse ground-glass opacities are noted, more prominently in the upper lobes of both lungs. Mediastinal structures cannot be evaluated optimally because contrast material is not given. Heart size is increased and minimal pericardial effusion is present. A port catheter tip projects over the right atrium. Calcific plaques are seen in the aortic walls. No obvious destructive bone lesions are seen within the scanned area.
Impression: Bilateral pleural effusion with compressive atelectasis; diffuse interstitial thickening and ground-glass opacities. Cardiomegaly with minimal pericardial effusion.

**SCALE-VLP prediction:**
Findings: Trachea and both main bronchi are patent. No obstructive lesion is noted. Diffuse interstitial thickening is present with patchy ground-glass densities. Dependent atelectasis is seen posteriorly. A moderate right and mild left pleural effusion is identified. Heart size is borderline enlarged. A right-sided port catheter tip overlies the right atrium. No definite destructive bone lesion is demonstrated. There is a small renal calculus in the mid right kidney.
Impression: Bilateral pleural effusions with associated atelectasis; diffuse interstitial lung disease pattern.

**M3D prediction:**
Findings: Trachea and both main bronchi are patent. Diffuse septal thickening and scattered ground-glass opacities are noted throughout both lungs. A moderate right pleural effusion and mild left pleural effusion with adjacent compressive atelectasis are present. Cardiomegaly is noted; the mediastinum is otherwise unremarkable. No aggressive osseous lesion is seen.
Impression: Bilateral pleural effusions with compressive atelectasis; diffuse interstitial changes.

**CT-CLIP prediction:**
Findings: Trachea and both main bronchi are patent. Diffuse interlobular septal thickening and areas of ground-glass opacity are seen in both lungs. A moderate right pleural effusion and mild left pleural effusion are present with associated atelectasis. No definite focal consolidation is seen. Heart size appears enlarged. No pathologically enlarged mediastinal lymph nodes are identified. No aggressive osseous lesion is demonstrated.
Impression: Bilateral pleural effusions with compressive atelectasis; diffuse interstitial thickening.

*[BLEU-4—ROUGE-1—METEOR]* **SCALE-VLP:**0.373—0.596—0.475 **M3D:**0.295—0.571—0.387
**CT-CLIP:**0.304—0.570—0.391

Figure 7: Qualitative comparison for Sample.

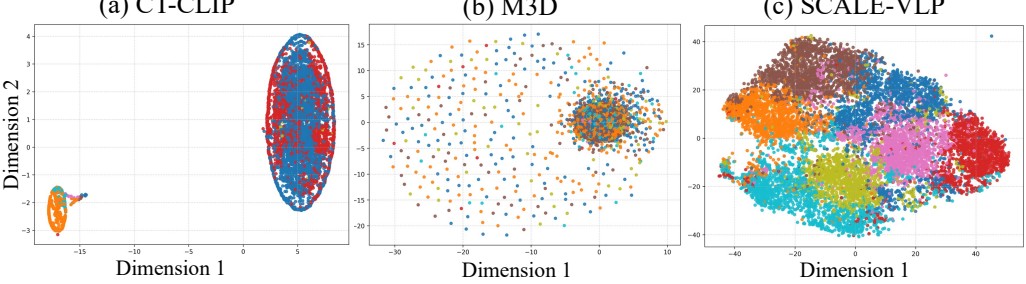

Figure 8: t-SNE of volume–report similarity.