# OpenReview forum: "SCALE-VLP: Soft-Weighted Contrastive Volumetric Vision–Language Pre-training with Spatial-Knowledge Semantics"
_ICLR.cc/2026/Conference — ICLR 2026 Conference Withdrawn Submission_

### Official Review · Reviewer_TmJw · 2025-10-16

**Soundness:** 2
**Presentation:** 2
**Contribution:** 1
**Rating:** 2
**Confidence:** 4

**Summary:**

This paper introduces SCALE-VLP, a soft-weighted contrastive vision-language pre-training framework tailored for 3D volumetric medical data, specifically CT scans and paired radiology reports. The method embeds volumetric spatial-coherent semantics and domain-specific clinical knowledge to enhance alignment, leveraging Soft-Weighted Contrastive Alignment (SWCA) objective that replaces binary supervision with dense, semantics-aware similarity matrices. SCALE-VLP is benchmarked for retrieval, classification, and report generation on large CT-report datasets, consistently outperforming existing models.

**Strengths:**

The paper proposes a single pretraining framework (SCALE-VLP) that supports three clinically relevant downstream tasks — report retrieval, report generation, and abnormality classification — using a shared encoder.

The method is benchmarked on both in-domain (CT-RATE) and out-of-domain (BIMCV-R) datasets, with detailed quantitative results across  retrieval, classification, and generation metrics.

The paper conducts ablation on both the spatial and knowledge weighting terms in SWCA.

**Weaknesses:**

1. The core motivation is confusing. The paper claims that "A central bottleneck lies in learning effective 3D representations that preserve the intrinsic spatial and semantic structure of the data". However this has already been explored in CT-CLIP, Merlin and fVLM which all use 3D backbones to encode volumetric information. As seen in Sec 3.1 model architecture, the authors also use 3D ViT to encode this information. The paper states "Each CT scan is paired with its radiology report, and both modalities are encoded using a 3D Vision Transformer and a medical-language encoder respectively". However, this modeling is not new and does not address an underexplored problem in medical imaging.

2. The proposed Soft-Weighted Contrastive Alignment (SWCA) objective is confusing to read. It seems like the authors want to propose a method to reweight a sigmoid contrastive objective using intra-modal similarities. How does this improve the "semantics" between CT and reports?

3. The authors claim SWCA "improves sample efficiency under limited supervision". There is no where in the paper that supports this claim.

4. The method heavily downsamples the input volume to 256 x 256 x 32. After patch embedding, the tokens along the axial dimension effective becomes 2 (given patch size of 16). How is it possible for the model to learn 3D “spatially coherent” distance? Furthermore, it is rather uncommon to only report the volume resolution without the actual spacing that the volume was resampeld to.

5. The “Medical knowledge fusion” is also overclaimed. This paper effectively embeds reports with a frozen medical LLM and computes a report-report similarity matrix to weight the loss. However, the image side receives no structured knowledge signal. In practice, this is a textual label-smoothing prior, not “knowledge fusion” between CT and text.

6. There is no analysis of potential bias or risk introduced by using an external LLM for "knowledge fusion".

7. The qualitative example in Figure 7 suggests a hallucinated finding (a renal calculus); can the authors provide a broader quantitative error analysis on hallucination or missing findings for report generation?

**Questions:**

Can the authors clarify, with empirical or theoretical evidence, why the spatial and knowledge priors are necessary?

How would SCALE-VLP perform under diverse domain shifts (e.g., different CT protocols or free-text reports from other healthcare systems)?

What are the training and inference costs for using SCALE-VLP?

---

> ### Author Response · Authors · 2025-11-27
> **Author Response (Part 1/5) - Clarifying Contribution**
>
> We thank the reviewer for the detailed feedback on our paper. We address each point in **Parts 1/5–5/5**. Kindly consider adjusting the score if our responses satisfactorily address your primary concerns. We would be happy to answer any further questions that may help support the acceptance of our work.
>
> > The core motivation is confusing. The paper claims that "A central bottleneck lies in learning effective 3D representations that preserve the intrinsic spatial and semantic structure of the data". However, this has already been explored in CT-CLIP, Merlin and fVLM which all use 3D backbones to encode volumetric information. As seen in Sec 3.1 model architecture, the authors also use 3D ViT to encode this information. The paper states "Each CT scan is paired with its radiology report, and both modalities are encoded using a 3D Vision Transformer and a medical-language encoder respectively". However, this modeling is not new and does not address an underexplored problem in medical imaging.
>
> We agree that using 3D backbones is not new; our contribution is on the alignment objective, not the encoder. CT-CLIP and Merlin both use 3D encoders but still adopt a “2D-style” contrastive loss: one global CT embedding vs. one global report embedding, trained with small batches and no explicit modelling of 3D geometry or local region–text relations. fVLM still relies on 2D CLIP-style image features and only achieves 3D-aware feature aggregation by exploiting additional anatomical segmentation annotations.
> SCALE-VLP keeps a similar 3D ViT + text backbone, but changes the objective: Soft-Weighted Contrastive Alignment with spatial and medical-knowledge weights makes the objective itself 3D- and semantics-aware, by weighting pairs according to volumetric kernels and medical-LLM similarities rather than treating each CT–report pair as a single global binary match. Our contribution is thus orthogonal to prior work: we build on their 3D encoders and target an underexplored aspect:  designing a contrastive objective intrinsically aware of volumetric spatial coherence (via 3D kernels over CT patches) and clinical semantics (via medical-LLM–based weights), and experiments demonstrate our effectiveness over CT-CLIP, Merlin, and fVLM.
>
> > The proposed Soft-Weighted Contrastive Alignment (SWCA) objective is confusing to read. It seems like the authors want to propose a method to reweight a sigmoid contrastive objective using intra-modal similarities. How does this improve the "semantics" between CT and reports?
>
> We thank the reviewer for their constructive feedback and will clarify the formulation in the text. Formally, SWCA is a SigLIP-style pairwise sigmoid loss, but with soft labels $w\_{ij} \in [0,1]$ instead of hard binary “Same VS Different” labels. These weights are constructed considering both (i) intra-modal CT-CT and text-text similarities and (ii) spatial and medical-knowledge kernels, so that $w_{ij}$ is high when CT$\_i$ and report$_j$ describe similar pathology/anatomy and near zero when they are clinically dissimilar. Next, we guide the CT-report similarity ($s\_{ij}$) toward the continuous semantic target weights $w\_{ij}$. Therefore, SWCA results in high similarity  $s\_{ij}$ if pathology and anatomy contain similar characteristics and vice-versa, resulting in growth of semantic alignment.
> On the other hand, under standard CLIP/SigLIP, CT$\_i$ is encouraged to be close only to its own report and to treat all other reports as equally negative, which is problematic in radiology, where many studies share partly related findings and features. SWCA instead encourages CT$\_i$ to be close to all semantically related reports (and vice versa) and strongly repels clearly mismatched pairs, yielding an embedding space whose geometry better reflects disease similarity and anatomical context. Empirically, we showed that SWCA leads to more structured CT-report similarity matrices and consistent gains in retrieval and other downstream tasks.

---

> ### Author Response · Authors · 2025-11-27
> **Author Response (Part 2/5) - Sample Efficiency, 3D Resolution, and Knowledge Fusion**
>
> > The authors claim SWCA "improves sample efficiency under limited supervision". There is no where in the paper that supports this claim.
>
> In SWCA, we move from a batch-wise softmax objective to a pairwise sigmoid formulation with soft labels, enabled by treating one modality encoder as fixed. This design (i) avoids batch-level normalization in the loss and reduces memory overhead for the similarity matrix, allowing a much larger effective batch size (220 CT–report pairs versus the smaller batches used by the CLIP-style baseline on the same 4×A40 setup), and therefore many more negatives per step; and (ii) replaces binary “same/different patient” supervision with continuous, structure-aware targets derived from CT–CT and text–text similarities. So, each paired example contributes “partial” supervision to many CT-report pairs instead of only one.  In  addition, in medical application with scarcity of paired data compared to general domain, SWCA allows information aggregation from multiple samples instead of only a single positive sample. Empirically, under the same CT-RATE data budget, Figure 2 and Tables 1 and 2 show that SCALE-VLP with SWCA substantially outperforms the CLIP-style base model across retrieval and generation metrics. Also, the steeper slope in Figure 2 as the number of samples increases further indicates that SWCA learns more efficiently as more data becomes available.
>
>
> > The method heavily downsamples the input volume to 256 x 256 x 32. After patch embedding, the tokens along the axial dimension effective becomes 2 (given patch size of 16). How is it possible for the model to learn 3D “spatially coherent” distance? Furthermore, it is rather uncommon to only report the volume resolution without the actual spacing that the volume was resampeld to.
>
> We thank the reviewer for pointing this out and apologize for the ambiguity. Our patch size is 16×16×4, following CT-ViT/M3D/CT-CLIP baselines, so a 256×256×32 volume yields 16×16×8 tokens, not 2 tokens along the axial axis. Thus, the 3D ViT attends over an 8-step craniocaudal grid, and each token covers ≈4 channels, which is sufficient to capture volumetric patterns (e.g., a nodule spanning several slices, basal-predominant disease, etc.). Moreover, the “spatially coherent distance” in SWCA is defined in physical 3D space: we resample CT-RATE to 0.75×0.75×1.5 mm spacing (as in CT-CLIP), and BIMCV-R to the same target spacing, then compute token centroids and pairwise 3D distances (in mm) to build the spatial kernel that shapes CT–CT and CT–text weights. In other words, SWCA explicitly preserves neighbourhood structure in a volumetric coordinate system rather than collapsing the scan into a flat bag of slices. We agree that reporting only 256×256×32 is incomplete and will revise Sec. 3 to state both resolution and voxel spacing for CT-RATE and BIMCV-R and to clarify the 16×16×4 patching scheme.
>
> > The “Medical knowledge fusion” is also overclaimed. This paper effectively embeds reports with a frozen medical LLM and computes a report-report similarity matrix to weight the loss. However, the image side receives no structured knowledge signal. In practice, this is a textual label-smoothing prior, not “knowledge fusion” between CT and text.
>
>
> We appreciate this concern. Scale-VLP mechanism is beyond simple textual label smoothing. In SWCA we first compute CT–CT and report–report similarity matrices, where the report side is augmented by a medical LLM and ontology embeddings. These intra-modal similarities are then combined into a cross-modal weight matrix $w_{ij}$​ that defines the soft targets for every CT-report pair in the loss. Therefore, the CT embeddings are explicitly optimized so that their geometry matches the knowledge-structured neighbourhood defined on the report side (and vice versa), rather than treating all non-matching reports as equally negative. This allows aggregating information from multiple scans for enhancing aligning reports (and vice versa). This yields CT representations that inherit clinically meaningful structure, providing an indirect but structured knowledge signal to the image side, rather than merely smoothing textual labels.

---

> ### Author Response · Authors · 2025-11-27
> **Author Response (Part 3/5) - Adding New External LLM Ablations**
>
> > There is no analysis of potential bias or risk introduced by using an external LLM for "knowledge fusion".
>
> We thank the reviewer for raising the risk concern regarding incorporating external LLMs for knowledge fusion. To explicitly evaluate this, we conducted additional ablation experiments using three different medical LLMs: HuatuoGPT (default), LLaMA3-Med42, and BioMistral. As shown in the below table, retrieval performance remains highly consistent across all three models for all pool sizes (N = 100–1564), with only marginal variations and all variants substantially outperform CT-CLIP, M3D, and Merlin. This consistency indicates that SCALE-VLP is not sensitive to any single LMM’s internal knowledge distribution, bias, or reasoning patterns. Instead, the LLM functions as a modular knowledge prior whose role is to provide coarse semantic guidance rather than dominate the alignment process. Practically, this means the framework is robust to the choice of external LLM and can safely integrate alternative models that meet institutional or regulatory requirements for safety, ethics, or bias control, without meaningful degradation in performance. Therefore, rather than introducing dependency or hidden bias from a specific LLM, the proposed knowledge fusion behaves as a plug-and-play component with stable behavior, substantially mitigating the risk of model-specific bias propagation in our setting.
>
>
> ## N = 100
> | N | Model | R@1 | R@5 | R@10 | R@50 | R@100 | SumR | R@1 | R@5 | R@10 | R@50 | R@100 | SumR |
> |-|-|-|-|-|-|-|-|-|-|-|-|-|-|
> | 100 | SCALE-VLP (HuatuoGPT)    | 13.0 | 40.0 | 56.0 | 94.0 | ---  | 203.0 | 14.0 | 42.0 | 59.0 | 93.0 | ---  | 208.0 |
> | 100 | SCALE-VLP (LLaMA3-Med42) | 11.0 | 37.0 | 55.0 | 97.0 | ---  | 200.0 | 11.0 | 34.0 | 58.0 | 94.0 | ---  | 197.0 |
> | 100 | SCALE-VLP (BioMistral)   | 15.0 | 37.0 | 53.0 | 97.0 | ---  | 202.0 | 15.0 | 35.0 | 55.0 | 96.0 | ---  | 201.0 |
>
>
> ---
>
> ## N = 500
> | N | Model | R@1 | R@5 | R@10 | R@50 | R@100 | SumR | R@1 | R@5 | R@10 | R@50 | R@100 | SumR |
> |-|-|-|-|-|-|-|-|-|-|-|-|-|-|
> | 500 | SCALE-VLP (HuatuoGPT)    | 3.4  | 12.2 | 19.8 | 53.2 | 72.6 | 161.2 | 4.4  | 11.8 | 21.0 | 51.6 | 72.6 | 161.4 |
> | 500 | SCALE-VLP (LLaMA3-Med42) | 3.6  | 10.6 | 18.6 | 55.0 | 73.0 | 160.8 | 3.0  | 11.0 | 19.2 | 53.4 | 73.4 | 160.0 |
> | 500 | SCALE-VLP (BioMistral)   | 3.4  | 11.0 | 18.2 | 53.4 | 71.4 | 157.4 | 4.0  | 13.4 | 20.2 | 52.6 | 71.2 | 161.4 |
>
>
> ---
>
> ## N = 1000
> | N | Model | R@1 | R@5 | R@10 | R@50 | R@100 | SumR | R@1 | R@5 | R@10 | R@50 | R@100 | SumR |
> |-|-|-|-|-|-|-|-|-|-|-|-|-|-|
> | 1000 | SCALE-VLP (HuatuoGPT)    | 1.8  | 6.4  | 11.6 | 35.8 | 51.9 | 107.5 | 2.2  | 6.5  | 11.5 | 35.8 | 51.2 | 107.2 |
> | 1000 | SCALE-VLP (LLaMA3-Med42) | 1.8  | 5.6  | 10.1 | 35.0 | 52.5 | 105.0 | 1.5  | 5.6  | 11.1 | 34.2 | 51.4 | 103.8 |
> | 1000 | SCALE-VLP (BioMistral)   | 1.6  | 5.6  | 10.0 | 34.1 | 52.5 | 103.8 | 1.9  | 6.8  | 10.7 | 34.8 | 51.6 | 105.8 |
>
>
> ## N = 1564
> | N | Model | R@1 | R@5 | R@10 | R@50 | R@100 | SumR | R@1 | R@5 | R@10 | R@50 | R@100 | SumR |
> |-|-|-|-|-|-|-|-|-|-|-|-|-|-|
> | 1564 | SCALE-VLP (HuatuoGPT)    | 1.2  | 4.2  | 7.4  | 25.1 | 38.8 | 76.7  | 1.3  | 4.2  | 7.4  | 25.4 | 38.5 | 76.8 |
> | 1564 | SCALE-VLP (LLaMA3-Med42) | 1.2  | 4.3  | 7.2  | 24.6 | 39.4 | 76.7  | 1.1  | 4.0  | 7.5  | 24.4 | 38.5 | 75.4 |
> | 1564 | SCALE-VLP (BioMistral)   | 0.9  | 3.8  | 7.0  | 23.8 | 39.1 | 74.5  | 1.2  | 3.9  | 6.8  | 24.7 | 40.0 | 76.6 |
>
> > The qualitative example in Figure 7 suggests a hallucinated finding (a renal calculus); can the authors provide a broader quantitative error analysis on hallucination or missing findings for report generation?
>
> We appreciate the reviewer’s attention to hallucination, which is indeed a critical issue for report generation. Unfortunately, CT-RATE does not provide exhaustive lesion-level or span-level annotations for all findings, so we currently lack the structured ground truth that would be required to compute a robust quantitative hallucination/missing-finding rate without conducting a dedicated radiologist reader study. Such a study is important but beyond the scope and timeline of this work.
> In the revision, we will explicitly discuss hallucination and missing findings as a key limitation of current CT–report benchmarks and models. We view a systematic, expert-annotated hallucination analysis and mitigation strategies as an important direction for future work.

---

> ### Author Response · Authors · 2025-11-27
> **Author Response (Part 4/5) - Added New Ablation on Spatial-only Variants**
>
> > Can the authors clarify, with empirical or theoretical evidence, why the spatial and knowledge priors are necessary?
>
> The role of the spatial and knowledge priors is supported both empirically and conceptually. Empirically, as shown in below table, removing spatial alignment (i.e., SWCA-base + Knowledge) reduces SumR by an average $5.6\%$ drop across pool sizes (IR and TR), while removing knowledge cues (i.e., SWCA-base + Spatial) demonstrates a lower average ($3.3\%$) drop compared to the full SWCA version. Finally, SWCA without both spatial and knowledge alignment (i.e., SWCA-base) yields the largest drop ($21.3\%$). The pattern is consistent from $N{=}100$ to $N{=}1564$, indicating that spatial reasoning is the most effective complementary contributor to cross-modal alignment, while medical knowledge provides additive gains. In addition, Table 2 reports analogous degradations in BLEU-4 (≈30% w/o Spatial, ≈47% w/o Spatial&Knowledge) and METEOR, and Table 3 shows consistent declines across all abnormality clusters when these priors are ablated. Qualitatively, the full model yields clear block-diagonal CT–report similarity matrices and pathology-aware clusters in Figures 3 and 8, whereas baselines without our priors exhibit weaker clustering and less interpretable structure. Conceptually, the spatial prior anchors alignment to volumetric continuity and organ-level neighborhoods, while the knowledge prior reshapes soft labels so that scans sharing ontology-level concepts (e.g., synonymic findings) are treated as partially positive instead of pure negatives. Together, these priors turn SWCA from a generic reweighting into a mechanism that injects 3D anatomy and radiological semantics into the loss, which is reflected in the consistent gains across retrieval, generation, and classification.
>
>
>
> ## $\text{N} = 100$
>
> | Model | IR R@1 | IR R@5 | IR R@10 | IR R@50 | IR R@100 | IR SumR | TR R@1 | TR R@5 | TR R@10 | TR R@50 | TR R@100 | TR SumR |
> |:-|:-|:-|:-|:-|:-|:-|:-|:-|:-|:-|:-|:-|
> | SCALE-VLP w/o Spatial \& Know. | 8.0 | 30.0 | 47.0 | 89.0 | – | 174.0 | 11.0 | 31.0 | 46.0 | 89.0 | – | 177.0 |
> | SCALE-VLP w/o Spatial | 12.0 | 37.0 | 55.0 | 93.0 | – | 197.0 | 13.0 | 40.0 | 54.0 | 92.0 | – | 199.0 |
> | SCALE-VLP w/o Knowledge | **17.0** | 36.0 | **56.0** | **96.0** | – | **205.0** | 10.0 | 37.0 | 58.0 | **94.0** | – | 199.0 |
> | **SCALE-VLP** | 13.0 | **40.0** | **56.0** | 94.0 | – | 203.0 | **14.0** | **42.0** | **59.0** | 93.0 | – | **208.0** |
>
> ---
>
> ## N = 500
>
> | Model | IR R@1 | IR R@5 | IR R@10 | IR R@50 | IR R@100 | IR SumR | TR R@1 | TR R@5 | TR R@10 | TR R@50 | TR R@100 | TR SumR |
> |:-|:-|:-|:-|:-|:-|:-|:-|:-|:-|:-|:-|:-|
> | SCALE-VLP w/o Spatial \& Know. | 3.0 | 8.2 | 14.2 | 46.6 | 66.8 | 138.8 | 2.8 | 9.2 | 15.2 | 45.4 | 64.4 | 137.0 |
> | SCALE-VLP w/o Spatial | 3.2 | 11.6 | 17.8 | 49.4 | 69.4 | 151.4 | 3.2 | 11.2 | 18.8 | 50.4 | 69.0 | 152.6 |
> | SCALE-VLP w/o Knowledge | 2.2 | 11.2 | **20.2** | 52.2 | 71.6 | 157.4 | 3.2 | 10.4 | 18.0 | 51.2 | 71.4 | 154.2 |
> | **SCALE-VLP** | **3.4** | **12.2** | 19.8 | **53.2** | **72.6** | **161.2** | **4.4** | **11.8** | **21.0** | **51.6** | **72.6** | **161.4** |
>
> ---
>
> ## N = 1000
>
> | Model | IR R@1 | IR R@5 | IR R@10 | IR R@50 | IR R@100 | IR SumR | TR R@1 | TR R@5 | TR R@10 | TR R@50 | TR R@100 | TR SumR |
> |:-|:-|:-|:-|:-|:-|:-|:-|:-|:-|:-|:-|:-|
> | SCALE-VLP w/o Spatial \& Know. | 1.5 | 5.0 | 8.5 | 28.3 | 46.0 | 89.3 | 1.3 | 5.5 | 9.2 | 26.9 | 44.6 | 87.5 |
> | SCALE-VLP w/o Spatial | 1.7 | 6.3 | 10.5 | 34.4 | 50.6 | 103.5 | 1.7 | 6.2 | 11.0 | 34.5 | 50.0 | 103.4 |
> | SCALE-VLP w/o Knowledge | 1.1 | 5.3 | 11.0 | 35.2 | **52.1** | 104.7 | 1.8 | 5.8 | 10.0 | 34.1 | **51.8** | 103.5 |
> | **SCALE-VLP** | **1.8** | **6.4** | **11.6** | **35.8** | 51.9 | **107.5** | **2.2** | **6.5** | **11.5** | **35.8** | 51.2 | **107.2** |
>
> ---
>
> ## $\text{N} = 1564$
>
> | Model | IR R@1 | IR R@5 | IR R@10 | IR R@50 | IR R@100 | IR SumR | TR R@1 | TR R@5 | TR R@10 | TR R@50 | TR R@100 | TR SumR |
> |:-|:-|:-|:-|:-|:-|:-|:-|:-|:-|:-|:-|:-|
> | SCALE-VLP w/o Spatial \& Know. | 1.0 | 3.7 | 6.1 | 20.5 | 33.1 | 64.4 | 1.0 | 3.6 | 6.0 | 20.4 | 32.0 | 63.0 |
> | SCALE-VLP w/o Spatial | 1.0 | **4.2** | 7.1 | 24.9 | 38.2 | 75.4 | 1.0 | 4.0 | 7.1 | 24.4 | 38.1 | 74.6 |
> | SCALE-VLP w/o Knowledge | 1.1 | 3.1 | 7.0 | **25.2** | **39.7** | 76.1 | 1.2 | 3.6 | 6.5 | 25.0 | **39.2** | 75.5 |
> | **SCALE-VLP** | **1.2** | **4.2** | **7.4** | 25.1 | 38.8 | **76.7** | **1.3** | **4.2** | **7.4** | **25.4** | 38.5 | **76.8** |

---

> ### Author Response · Authors · 2025-11-27
> **Author Response (Part 5/5) - Domain Shift Robustness and Computational Cost**
>
> > How would SCALE-VLP perform under diverse domain shifts (e.g., different CT protocols or free-text reports from other healthcare systems)?
>
> We agree that robustness under domain shift is important. In practice, SCALE-VLP is already trained and evaluated under heterogeneous imaging and reporting conditions. CT-RATE itself includes substantial variation in reconstruction methods and acquisition parameters, and our splits are strictly patient-wise, so the in-domain test set already reflects protocol variability. More importantly, Sec. 4.3 and Table 4 report zero-shot results on BIMCV-R, an external dataset curated from a different healthcare system with distinct scanners, reconstruction pipelines, and free-text reports (originally in Spanish and normalized to English). They also cover distinct body organs, and have has different patient demographics. The table below summarizes the main differences between these two datasets. This setting constitutes a genuine domain shift and therefore a meaningful test of cross-domain robustness. We train only on CT-RATE and freeze all parameters for BIMCV-R, yet SCALE-VLP consistently matches or outperforms CT-CLIP, Merlin, and fVLM across retrieval (N=100, 1000) and report-generation metrics. This suggests that the SWCA objective and knowledge/spatial weighting yield representations that transfer well across different CT protocols and reporting styles, although a broader survey of sites and modalities is a valuable direction for future work.
>
>
>
> | | **CT-RATE** | **BIMCV-R** |
> |---|---|---|
> | **Coverage** | Non-contrast **chest** CT only | **Multi-region/Whole-body** CT |
> | **Hospitals** | Single centre (Istanbul Medipol Univ.) | Multi-centre BIMCV (~11, Valencian Region) |
> | **Vendors** | Philips 61.5%, Siemens 30.1%, PNMS 8.4% | Multiple (not detailed) |
> | **Resolution** | 512, 768, 1024 px | Variable; typically $\geq$512 px |
> | **# of Slices** | 100-600 (avg 305) | 101–670 (avg 279) |
>
>
>
>
>
>  > What are the training and inference costs for using SCALE-VLP?
>
>
> Regarding computational cost, SWCA only modifies the loss, not the encoder stack, so there are no extra forward passes compared to CT-CLIP/M3D at inference and  embeddings for soft weights can be saved once to avoid multiple forward paths during all training steps. All experiments are run on 4×A40 (48GB) with bf16, effective batch size 220 (55 CT–report pairs/GPU), AdamW + cosine schedule and no gradient accumulation. Under this setup, pre-training for 10 epochs takes 3.8 hours for SCALE-VLP, whereas CT-CLIP, M3D, and Merlin require 5.3 h, 5.58 h, and 10 h, respectively, under identical settings. Fine-tuning SCALE-VLP for report generation for 4 epochs adds about 3 hours. At inference time, SWCA and the spatial/knowledge kernels are not used: SCALE-VLP behaves as a standard dual encoder with a 3D ViT and text encoder, so a CT–report similarity requires one forward pass through each encoder plus a dot product, with the same theoretical FLOPs and practical latency as CT-CLIP/M3D for the same backbone and sequence lengths (CT embeddings for retrieval can be precomputed once and reused).

---

### Official Review · Reviewer_BXEP · 2025-10-20

**Soundness:** 2
**Presentation:** 2
**Contribution:** 3
**Rating:** 4
**Confidence:** 3

**Summary:**

SCALE-VLP introduces volumetric spatial information and medical knowledges into vision-language pretraining through a soft contrastive objective. Specifically, the weights of the soft objective are defined through a volumetric kernel that encodes spatial structure, while medical knowledge is obtained through embeddings of a pretrained Medical VLM. The authors evaluate across multiple downstream tasks such as cross-modal retrieval, report generation, abnormality classification, and zero-shot generalization under domain shift.

**Strengths:**

- This seems like a novel extension of soft contrastive learning to an important problem of vision language alignment for CT VLMs.
- Performance in cross-modal retrieval and Report generation seems to improve

**Weaknesses:**

- **Unclear Method description**:
    - Line 179: Unclear where $z_i$ came from. Is i indexing a volume patch or an entire sample?
    - Equation 6: What score is computed to get $r_{i,m}$? What are the inputs and outputs of this score?
    - Equation 8: Since the two exponential terms likely differ in scale, their direct multiplication may be dominated by one component. How do the authors determine $\kappa_\mu$ and $\kappa)_\Sigma$ hyperparameters to prevent one component from dominating? Shouldn’t these parameters be set dynamically during training as $\mu$ and $\Sigma$ get updated?
    - Can the authors clarify what “pool size” (line 312) is referring to? What dimension are we pooling over? And why does increasing it lead to a decrease in performance in table 1? How would a practitioner choose this hyperparameter?
- **Clarity could be significantly improved**:
    - In line 58-60, the authors argue that the problem with fVLM is the use of discrete masks, but do not indicate why this is a limitation. Can the authors clarify what causes discrete masks a poor modeling choice? And furthermore clarify what type of “continuous structure” in volumetric data is available to exploit? What happens to the learned representation when using discrete mask during contrastive pretraining and why is this unfavorable?
    - Can the authors also point to specific examples of what “fine-grained spatial details” is referring to (line 56)?
    - Line 134: The authors claim: “current frameworks fall short in considering spatially grounded and clinically guided semantic alignment”. I’m not sure I understand where this claim is referring to. Can you strengthen this argument and point to specific shortcomings? What is the algorithmic limitation that causes these models to fall short?
    - Line 314-315: The authors claim “SCALE-VLP surpasses it, suggesting that our spatial– and knowledge–aware alignment effectively capture fine-grained clinical semantics.” However, it is unclear to me the connection between improved performance in cross-modal retrieval and the impact of spatial/knowledge-aware alignment on fine-grained clinical semantics. Can the authors state this connection more clearly? Since the knowledge-alignment depends on a global embedding of the entire report, I’m not sure how this can possibly lead to fine-grained semantics.
- The current ablation study is missing "SCALE-VLP w/o knowledge". Would it be possible to include this ablation so that we can understand the individual contribution of the spatial alignment?
- In the experiments, I dont understand what is the learning objective is for “SCALE-VLP w/o spatial and knowledge”. If the full pretraining objective is defined by equation 11, what is left in the learning objective after removing both spatial and knowledge terms?
- In table 3: Its not clear to me that ScaleVLP does better in abnormality classification. Specifically, it seems like fVLM has stronger cluster-average accuracy. Can the authors clarify why a practitioner would prefer ScaleVLP over an existing method in abnormality classification?

Minor
- In conclusion: “iii)” and “ii)” seem to come out of order
- citations should be \citep instead of \cite when used as references. Otherwise, the citations read as part of the sentence.

**Questions:**

Please refer to weaknesses for the majority of the questions.
- How sensitive is the method to the Medical-Knowledge VLM? Can the authors test on multiple Medical VLMs to show that the method generalizes beyond a single backbone?
- Why is the 3d vision encoder frozen? Why wouldn't it be more effective to jointly optimize the learned features end-to-end? If its about limited compute requirements, why not use LORA for the encoder?

---

> ### Author Response · Authors · 2025-11-27
> **Author Response (Part 1/3) - Clarifying Method Description**
>
> We thank the reviewer for providing detailed feedback on our paper. We have addressed all concerns as follows **(Part 1/3 – Part 3/3)**. Kindly let us know if this resolves your concerns.
>
> >Unclear Method description:
> > * Line 179: Unclear where  came from. Is i indexing a volume patch or an entire sample?
> > * Equation 6: What score is computed to get ? What are the inputs and outputs of this score?
> >* Equation 8: Since the two exponential terms likely differ in scale, their direct multiplication may be dominated by one component.
> >* How do the authors determine  and  hyperparameters to prevent one component from dominating? Shouldn’t these parameters be set dynamically during training as  and  get updated?
> >* Can the authors clarify what “pool size” (line 312) is referring to? What dimension are we pooling over? And why does increasing it lead to a decrease in performance in table 1?..
>
> We thank the reviewer for pointing out these clarity issues and will clarify accordingly. In Eq (6), the index $i$ refers to the $i$-th CT volume and $z\_i$ refers to the encoder embedding for the i-th patient, while $m$ indexes the 3D patches within that volume. For each patch, we compute a non-negative saliency score from its embedding (using the $\ell_2$ norm of the final-layer patch feature, which we will make explicit) and normalize it to obtain the attention weights, from which the weighted centroid and covariance used in Eq (7) are derived. In Eq (8), the spatial kernel combines both spatial location and spread, with $\kappa_\mu$ and $\kappa_\Sigma$ acting as fixed scale parameters chosen to keep the two distance terms numerically comparable. These values are set based on physical CT spacing and embedding scale, specifically using the empirical standard deviations of the corresponding distance measures on a subset of the training data, preventing either term from dominating. In practice, moderate variations in these parameters have a much smaller effect than enabling or disabling spatial/knowledge weighting. The pool size N is the number of candidate CT-report pairs evaluated during retrieval (1 correct pair + N−1 samples). It only affects evaluation, not training. As N increases, the retrieval task becomes harder, which naturally lowers the retrieval scores.
>
> > Clarity could be significantly improved:
> > * In line 58-60, the authors argue that the problem with fVLM is the use of discrete masks, but do not indicate why this is a limitation..
> > * Can the authors also point to specific examples of what “fine-grained spatial details” is referring to (line 56)?
> > * Line 134: The authors claim: “current frameworks fall short in considering spatially grounded and clinically guided semantic alignment”..
> > * Line 314-315: The authors claim “SCALE-VLP surpasses it, suggesting that our spatial– and knowledge–aware alignment effectively capture fine-grained clinical semantics.”..
>
> Our remark about fVLM’s discrete masks is that supervision is applied only at the organ-mask level, where each segmented anatomy is treated as a single region with one embedding and a hard label. This is restrictive for volumetric CT, where many findings vary continuously in 3D and span across or within masks (e.g., nodules across slices or peripheral vs. central consolidation).  With purely discrete masks, the loss has no incentive to distinguish such fine-grained spatial details within an organ and cannot capture such intra-organ spatial variation; in contrast, SCALE-VLP uses continuous 3D centroids and spreads of patch clusters, tying supervision to smooth volumetric neighbourhoods rather than segmentation boundaries. In addition, Compared to SCALE-VLP that only relies on the input report and CT-scans, fVLM relies on additive information from organ masked alongside CT-scans which may limit affect its applicability and may be influenced by the quality of the mask.
> By “current frameworks fall short,” we refer to two limitations: (i) CT-CLIP/Merlin use global binary alignment (one positive pair vs all others equally negative) that ignores spatial structure and cross-patient semantic similarity, and (ii) fVLM performs mask-local but still binary alignment without volumetric continuity or soft cross-patient similarities. SCALE-VLP instead learns a structured target kernel combining 3D spatial proximity and ontology/LLM-based concept similarity.
> Finally, by “fine-grained clinical semantics”,We are not claiming phrase-to-voxel grounding; rather, spatial/knowledge-aware weighting leads the embedding geometry to separate more nuanced disease and abnormality patterns instead of only normal vs. abnormal. This is supported by (i) improved retrieval at large pool sizes and per-cluster abnormality retrieval/classification, and (ii) clearer pathology-aligned block structure in the CT–report similarity matrices of Fig. 3 compared to baselines. We will revise the wording to emphasize this representation-level effect and avoid over-claiming phrase-level grounding.

---

> ### Author Response · Authors · 2025-11-27
> **Author Response (Part 2/3) - Added New Ablation Experiment**
>
> > The current ablation study is missing "SCALE-VLP w/o knowledge". Would it be possible to include this ablation so that we can understand the individual contribution of the spatial alignment?
>
> We conducted an **additional ablation experiment**, and the results are presented in the updated table below; Both priors contribute substantial gains over the no-prior baseline, but the spatial prior consistently yields slightly larger improvements. For retrieval, the spatial-only variant (w/o Medical) improves IR SumR over the base model by +31.0, +18.6, +15.4, and +11.7 for N=100, 500, 1000, and 1564, whereas the knowledge-only variant (w/o Spatial) yields +23.0, +12.6, +14.2, and +11.0, with a similar pattern on TR SumR. In Tables 2 and 3 (report generation and abnormality classification), ablating spatial also leads to larger drops than ablating knowledge, indicating that spatial coherence is particularly important for downstream tasks. The full SCALE-VLP model, which combines both priors, consistently achieves the best performance. We revised Sec. 4.1.1 to replace “dominant contributor” with “most effective complementary contributor” while their combination yields the largest overall gains. **We have also updated Table 1 in the paper to reflect the new ablation.**
>
>
> ## $\text{N} = 100$
>
> | Model | IR R@1 | IR R@5 | IR R@10 | IR R@50 | IR R@100 | IR SumR | TR R@1 | TR R@5 | TR R@10 | TR R@50 | TR R@100 | TR SumR |
> |:-|:-|:-|:-|:-|:-|:-|:-|:-|:-|:-|:-|:-|
> | SCALE-VLP w/o Spatial \& Know. | 8.0 | 30.0 | 47.0 | 89.0 | – | 174.0 | 11.0 | 31.0 | 46.0 | 89.0 | – | 177.0 |
> | SCALE-VLP w/o Spatial | 12.0 | 37.0 | 55.0 | 93.0 | – | 197.0 | 13.0 | 40.0 | 54.0 | 92.0 | – | 199.0 |
> | SCALE-VLP w/o Knowledge | **17.0** | 36.0 | **56.0** | **96.0** | – | **205.0** | 10.0 | 37.0 | 58.0 | **94.0** | – | 199.0 |
> | **SCALE-VLP** | 13.0 | **40.0** | **56.0** | 94.0 | – | 203.0 | **14.0** | **42.0** | **59.0** | 93.0 | – | **208.0** |
>
> ---
>
> ## $\text{N} = 500$
>
> | Model | IR R@1 | IR R@5 | IR R@10 | IR R@50 | IR R@100 | IR SumR | TR R@1 | TR R@5 | TR R@10 | TR R@50 | TR R@100 | TR SumR |
> |:-|:-|:-|:-|:-|:-|:-|:-|:-|:-|:-|:-|:-|
> | SCALE-VLP w/o Spatial \& Know. | 3.0 | 8.2 | 14.2 | 46.6 | 66.8 | 138.8 | 2.8 | 9.2 | 15.2 | 45.4 | 64.4 | 137.0 |
> | SCALE-VLP w/o Spatial | 3.2 | 11.6 | 17.8 | 49.4 | 69.4 | 151.4 | 3.2 | 11.2 | 18.8 | 50.4 | 69.0 | 152.6 |
> | SCALE-VLP w/o Knowledge | 2.2 | 11.2 | **20.2** | 52.2 | 71.6 | 157.4 | 3.2 | 10.4 | 18.0 | 51.2 | 71.4 | 154.2 |
> | **SCALE-VLP** | **3.4** | **12.2** | 19.8 | **53.2** | **72.6** | **161.2** | **4.4** | **11.8** | **21.0** | **51.6** | **72.6** | **161.4** |
>
> ---
>
> ## $\text{N} = 1000$
>
> | Model | IR R@1 | IR R@5 | IR R@10 | IR R@50 | IR R@100 | IR SumR | TR R@1 | TR R@5 | TR R@10 | TR R@50 | TR R@100 | TR SumR |
> |:-|:-|:-|:-|:-|:-|:-|:-|:-|:-|:-|:-|:-|
> | SCALE-VLP w/o Spatial \& Know. | 1.5 | 5.0 | 8.5 | 28.3 | 46.0 | 89.3 | 1.3 | 5.5 | 9.2 | 26.9 | 44.6 | 87.5 |
> | SCALE-VLP w/o Spatial | 1.7 | 6.3 | 10.5 | 34.4 | 50.6 | 103.5 | 1.7 | 6.2 | 11.0 | 34.5 | 50.0 | 103.4 |
> | SCALE-VLP w/o Knowledge | 1.1 | 5.3 | 11.0 | 35.2 | **52.1** | 104.7 | 1.8 | 5.8 | 10.0 | 34.1 | **51.8** | 103.5 |
> | **SCALE-VLP** | **1.8** | **6.4** | **11.6** | **35.8** | 51.9 | **107.5** | **2.2** | **6.5** | **11.5** | **35.8** | 51.2 | **107.2** |
>
> ---
>
> ## $\text{N} = 1564$
>
> | Model | IR R@1 | IR R@5 | IR R@10 | IR R@50 | IR R@100 | IR SumR | TR R@1 | TR R@5 | TR R@10 | TR R@50 | TR R@100 | TR SumR |
> |:-|:-|:-|:-|:-|:-|:-|:-|:-|:-|:-|:-|:-|
> | SCALE-VLP w/o Spatial \& Know. | 1.0 | 3.7 | 6.1 | 20.5 | 33.1 | 64.4 | 1.0 | 3.6 | 6.0 | 20.4 | 32.0 | 63.0 |
> | SCALE-VLP w/o Spatial | 1.0 | **4.2** | 7.1 | 24.9 | 38.2 | 75.4 | 1.0 | 4.0 | 7.1 | 24.4 | 38.1 | 74.6 |
> | SCALE-VLP w/o Knowledge | 1.1 | 3.1 | 7.0 | **25.2** | **39.7** | 76.1 | 1.2 | 3.6 | 6.5 | 25.0 | **39.2** | 75.5 |
> | **SCALE-VLP** | **1.2** | **4.2** | **7.4** | 25.1 | 38.8 | **76.7** | **1.3** | **4.2** | **7.4** | **25.4** | 38.5 | **76.8** |
>
>
>
>
> > In the experiments, I don't understand what is the learning objective is for “SCALE-VLP w/o spatial and knowledge”. If the full pretraining objective is defined by equation 11, what is left in the learning objective after removing both spatial and knowledge terms?
>
> Equation (11) defines the general SWCA objective as a contrastive loss with soft targets $w\_{ij}$ that incorporate spatial and knowledge-based weighting. In the SCALE-VLP w/o spatial \& knowledge variant, we do not remove the loss itself; instead, we set $w\_{ij}$ based on  CT encoder cls token similarities and the knowledge side will be ignored. The other ablations then replace these soft targets derived from (i) spatial kernel only or (ii) knowledge information only, making it clear what each component adds on top of this base objective.

---

> ### Author Response · Authors · 2025-11-27
> **Author Response (Part 3/3) - Adding New External LLM Ablations**
>
> > In table 3: Its not clear to me that ScaleVLP does better in abnormality classification. Specifically, it seems like fVLM has stronger cluster-average accuracy. Can the authors clarify why a practitioner would prefer ScaleVLP over an existing method in abnormality classification?
>
> SCALE-VLP is more practical and scalable for real-world clinical deployment because it removes the need for manually annotated anatomy masks or region-level annotations which fVLM requires. In other words, fVLM inputs are richer, but also less practical. SCALE-VLP operates directly on raw 3D CT volumes and reports, enabling end-to-end training with minimal preprocessing. In addition, while SCALE-VLP does not outperform fVLM on every cluster, it achieves higher overall performance and more balanced behavior across clusters. While fVLM excels on some families and performs poorly on others, SCALE-VLP trades small losses on the strongest clusters for substantial gains on the weakest ones. This results in higher cluster-average accuracy and more reliable performance across abnormality types, which is clinically preferable.
>
>
> > In conclusion: “iii)” and “ii)” seem to come out of order
> > citations should be \citep instead of \cite when used as references. Otherwise, the citations read as part of the sentence. ->
>
> We thank the reviewer for pointing out these minor editorial errors, and fixed these minor edits in the text
>
> > How sensitive is the method to the Medical-Knowledge VLM? Can the authors test on multiple Medical VLMs to show that the method generalizes beyond a single backbone?
>
> We thank the reviewer and have added an experiment where SCALE-VLP is trained with two other medical VLMs: LLaMA3-Med42, and BioMistral. HuatuoGPT is our default. As shown in the table below, retrieval performance is very similar across all three, and all variants remain clearly stronger than CT-CLIP/M3D/Merlin. This indicates that SCALE-VLP is not sensitive to a specific medical VLM; the knowledge term is plug-and-play, with HuatuoGPT chosen only because it gives slightly better average performance.
>
> ## N = 100
> | Model | R@1 | R@5 | R@10 | R@50 | R@100 | SumR | R@1 | R@5 | R@10 | R@50 | R@100 | SumR |
> |-|-|-|-|-|-|-|-|-|-|-|-|-|
> | SCALE-VLP (HuatuoGPT)    | 13.0 | 40.0 | 56.0 | 94.0 | ---  | 203.0 | 14.0 | 42.0 | 59.0 | 93.0 | ---  | 208.0 |
> | SCALE-VLP (LLaMA3-Med42) | 11.0 | 37.0 | 55.0 | 97.0 | ---  | 200.0 | 11.0 | 34.0 | 58.0 | 94.0 | ---  | 197.0 |
> | SCALE-VLP (BioMistral)   | 15.0 | 37.0 | 53.0 | 97.0 | ---  | 202.0 | 15.0 | 35.0 | 55.0 | 96.0 | ---  | 201.0 |
>
>
> ## N = 500
> | Model | R@1 | R@5 | R@10 | R@50 | R@100 | SumR | R@1 | R@5 | R@10 | R@50 | R@100 | SumR |
> |-|-|-|-|-|-|-|-|-|-|-|-|-|
> | SCALE-VLP (HuatuoGPT)    | 3.4  | 12.2 | 19.8 | 53.2 | 72.6 | 161.2 | 4.4  | 11.8 | 21.0 | 51.6 | 72.6 | 161.4 |
> | SCALE-VLP (LLaMA3-Med42) | 3.6  | 10.6 | 18.6 | 55.0 | 73.0 | 160.8 | 3.0  | 11.0 | 19.2 | 53.4 | 73.4 | 160.0 |
> | SCALE-VLP (BioMistral)   | 3.4  | 11.0 | 18.2 | 53.4 | 71.4 | 157.4 | 4.0  | 13.4 | 20.2 | 52.6 | 71.2 | 161.4 |
>
> ## N = 1000
> | Model | R@1 | R@5 | R@10 | R@50 | R@100 | SumR | R@1 | R@5 | R@10 | R@50 | R@100 | SumR |
> |-|-|-|-|-|-|-|-|-|-|-|-|-|
> | SCALE-VLP (HuatuoGPT)    | 1.8  | 6.4  | 11.6 | 35.8 | 51.9 | 107.5 | 2.2  | 6.5  | 11.5 | 35.8 | 51.2 | 107.2 |
> | SCALE-VLP (LLaMA3-Med42) | 1.8  | 5.6  | 10.1 | 35.0 | 52.5 | 105.0 | 1.5  | 5.6  | 11.1 | 34.2 | 51.4 | 103.8 |
> | SCALE-VLP (BioMistral)   | 1.6  | 5.6  | 10.0 | 34.1 | 52.5 | 103.8 | 1.9  | 6.8  | 10.7 | 34.8 | 51.6 | 105.8 |
>
>
> ## N = 1564
> | Model | R@1 | R@5 | R@10 | R@50 | R@100 | SumR | R@1 | R@5 | R@10 | R@50 | R@100 | SumR |
> |-|-|-|-|-|-|-|-|-|-|-|-|-|
> | SCALE-VLP (HuatuoGPT)    | 1.2  | 4.2  | 7.4  | 25.1 | 38.8 | 76.7  | 1.3  | 4.2  | 7.4  | 25.4 | 38.5 | 76.8 |
> | SCALE-VLP (LLaMA3-Med42) | 1.2  | 4.3  | 7.2  | 24.6 | 39.4 | 76.7  | 1.1  | 4.0  | 7.5  | 24.4 | 38.5 | 75.4 |
> | SCALE-VLP (BioMistral)   | 0.9  | 3.8  | 7.0  | 23.8 | 39.1 | 74.5  | 1.2  | 3.9  | 6.8  | 24.7 | 40.0 | 76.6 |
>
>
> >Why is the 3d vision encoder frozen? Why wouldn't it be more effective to jointly optimize the learned features end-to-end? If its about limited compute requirements, why not use LORA for the encoder?
>
> We freeze the 3D vision encoder for two reasons. First, following LiT (Zhai et al., CVPR 2022), which shows that keeping a strong pre-trained vision encoder fixed and only learning the alignment improves zero-shot transfer and stabilizes contrastive training, we treat the 3D ViT as a fixed CT backbone and focus on improving the objective (SWCA), making the source of gains clearer. Second, full end-to-end fine-tuning of a large 3D ViT is highly compute-intensive and did not show clear benefits in our preliminary experiments, particularly for zero-shot BIMCV-R. Incorporating LoRA-style lightweight adaptation could indeed be useful, but is outside of the current scope of the paper; we leave this to future work.

---

### Official Review · Reviewer_GLiq · 2025-10-27

**Soundness:** 3
**Presentation:** 3
**Contribution:** 2
**Rating:** 4
**Confidence:** 4

**Summary:**

This paper introduces SCALE-VLP, a volumetric vision–language pre-training framework for CT–report alignment. The authors propose a Soft-Weighted Contrastive Alignment (SWCA) loss that models continuous (rather than binary) image–text similarities, and integrate two additional weighting schemes: (i) a spatially coherent semantic alignment derived from 3D geometry, and (ii) a knowledge-infused semantic alignment based on medical large language models (LLMs). The method aims to improve sample efficiency and enable robust cross-task and cross-domain transfer. Experiments on CT-RATE and BIMCV-R datasets show strong performance on retrieval, report generation, and classification.

**Strengths:**

- The paper addresses an important and underexplored problem — volumetric (3D) vision–language pre-training for medical imaging.
- The idea of soft-weighted alignment for partial image–report correspondence is reasonable and clearly explained.
- Empirical results are extensive, covering multiple tasks (retrieval, report generation, classification) and datasets (CT-RATE, BIMCV-R).
- The writing is clear, and the overall system is well-structured.

**Weaknesses:**

The proposed Soft-Weighted Contrastive Alignment is a modest variant of prior sigmoid-based contrastive objectives (e.g., SigLIP, CWCL), extended with heuristic spatial and knowledge-based weighting. These additions are conceptually straightforward and lack clear theoretical justification or novel insight into vision–language alignment.

The methodological design is largely empirical, with multiple hyperparameters (κµ, κΣ, α) introduced without sensitivity or ablation analysis. The “knowledge-infused” component is loosely motivated and functions mainly as a text-similarity reweighting step, without evidence of genuine knowledge transfer or clinical reasoning benefit.

The experimental evaluation, though extensive, remains narrow in scope and rigor. All tests are CT-based, limiting the claimed cross-domain generalization. Reported gains are modest and lack statistical validation. Qualitative results are shallow, with no expert verification to support the claimed clinical plausibility.

Overall, the work is a careful system extension rather than a substantive methodological advance. While performance improvements are consistent, the contribution is incremental and lacks the conceptual depth expected for ICLR.

**Questions:**

Please refer to the Weaknesses section.

**I am willing to raise my score according to the rebuttal.**

---

> ### Author Response · Authors · 2025-11-25
> **Author Response (Part 1/2) - Clarifying Novelty of SCALE-VLP + Adding New Baseline**
>
> We thank you for a detailed and thorough assessment of this work. We have addressed all concerns as follows **(Part 1/2)**.
> Concerns 2, 3 and 4 are addressed in **Part 2/2**. Please let us know if there are any additional points we can discuss, to improve your assessment of our work.
>
> >W1. The proposed Soft-Weighted Contrastive Alignment is a modest variant of prior sigmoid-based contrastive objectives (e.g., SigLIP, CWCL), extended with heuristic spatial and knowledge-based weighting. These additions are conceptually straightforward and lack clear theoretical justification or novel insight into vision–language alignment.
>
>
>  We agree that SWCA is, at the objective level, a SigLIP-style pairwise sigmoid loss. Our contribution lies in the construction of soft labels $\(w_{ij} \in [0,1]\)$ for 3D medical CT, derived jointly from (i) intra-modal CT–CT and report–report similarities and (ii) domain-specific spatial and medical-knowledge kernels, so each CT is attracted to clinically related reports and repelled from clearly mismatched ones. CWCL-style losses do not combine both 3D spatial structure and medical priors in this way, nor have they been instantiated for volumetric CT–text alignment. This design is particularly important in radiology, where many “negatives” in CLIP/SigLIP are only partially dissimilar and treating them as equally negative harms the geometry of the embedding space.
>
> In line with the reviewer’s suggestion, **we included a CWCL-style as an additional baseline** that uses a continuous weighting but no combined spatial and knowledge kernels. **Full results are in updated Table 1**. This model achieves SumR scores of 152/162 (IR/TR) for N=100, 108.2/110.6 for N=500, 70.1/71.7 for N=1000, and 49.8/51.3 for N=1564, consistently underperforming the full SWCA configuration. This confirms that the gains of SCALE-VLP are not merely due to generic continuous weighting, but stem from the proposed spatial- and knowledge-driven construction of $w_{ij}$
>
>
> ## $\text{N} = 100$
>
> | Model | IR R@1 | IR R@5 | IR R@10 | IR R@50 | IR R@100 | IR SumR | TR R@1 | TR R@5 | TR R@10 | TR R@50 | TR R@100 | TR SumR |
> |:-|:-|:-|:-|:-|:-|:-|:-|:-|:-|:-|:-|:-|
> | Baseline (CWCL-style) | 6.0 | 27.0 | 40.0 | 79.0 | – | 152.0 | 9.0 | 29.0 | 44.0 | 80.0 | – | 162.0 |
> | **SCALE-VLP (Ours)** | **13.0** | **40.0** | **56.0** | **94.0** | – | **203.0** | **14.0** | **42.0** | **59.0** | **93.0** | – | **208.0** |
>
> ---
>
> ## $\text{N} = 500$
>
> | Model | IR R@1 | IR R@5 | IR R@10 | IR R@50 | IR R@100 | IR SumR | TR R@1 | TR R@5 | TR R@10 | TR R@50 | TR R@100 | TR SumR |
> |:-|:-|:-|:-|:-|:-|:-|:-|:-|:-|:-|:-|:-|
> | Baseline (CWCL-style) | 2.6 | 6.2 | 12.4 | 34.2 | 52.8 | 108.2 | 2.6 | 7.8 | 11.4 | 35.4 | 53.4 | 110.6 |
> | **SCALE-VLP (Ours)** | **3.4** | **12.2** | **19.8** | **53.2** | **72.6** | **161.2** | **4.4** | **11.8** | **21.0** | **51.6** | **72.6** | **161.4** |
>
> ---
>
> ## $\text{N} = 1000$
>
> | Model | IR R@1 | IR R@5 | IR R@10 | IR R@50 | IR R@100 | IR SumR | TR R@1 | TR R@5 | TR R@10 | TR R@50 | TR R@100 | TR SumR |
> |:-|:-|:-|:-|:-|:-|:-|:-|:-|:-|:-|:-|:-|
> | Baseline (CWCL-style) | 1.1 | 4.1 | 6.9 | 23.0 | 35.0 | 70.1 | 1.1 | 4.7 | 7.4 | 23.1 | 35.4 | 71.7 |
> | **SCALE-VLP (Ours)** | **1.8** | **6.4** | **11.6** | **35.8** | **51.9** | **107.5** | **2.2** | **6.5** | **11.5** | **35.8** | **51.2** | **107.2** |
>
> ---
>
> ## $\text{N} = 1564$
>
> | Model | IR R@1 | IR R@5 | IR R@10 | IR R@50 | IR R@100 | IR SumR | TR R@1 | TR R@5 | TR R@10 | TR R@50 | TR R@100 | TR SumR |
> |:---|:---|:---|:---|:---|:---|:---|:---|:---|:---|:---|:---|:---|
> | Baseline (CWCL-style) | 0.6 | 3.1 | 4.5 | 16.1 | 25.5 | 49.8 | 0.6 | 3.0 | 5.0 | 17.1 | 25.6 | 51.3 |
> | **SCALE-VLP (Ours)** | **1.2** | **4.2** | **7.4** | **25.1** | **38.8** | **76.7** | **1.3** | **4.2** | **7.4** | **25.4** | **38.5** | **76.8** |

---

> ### Author Response · Authors · 2025-11-25
> **Author Response (Part 2/2) - Addressing Hyperparameter Design, Evaluation Scope and Conceptual Contribution**
>
> >The methodological design is largely empirical, with multiple hyperparameters (κµ, κΣ, α) introduced without sensitivity or ablation analysis...
>
>  We thank the reviewer and note that an ablation on α is already available in the Appendix (A.4). The mixing coefficient α is not arbitrary: Fig. 5-6 and Tables. 7-10 demonstrate a sweep over α, showing that performance is stable over a broad range and peaks around α = 0.5, which we then fix for all main experiments. The kernel scales $κ\_{\mu}$, $κ\_{\Sigma}$ are set from the physical CT spacing and concept-embedding scale and are not tuned per task; in practice, moderate changes to these values have a much smaller effect than turning spatial/knowledge weighting on or off, and we made this explanation explicit in the revised paper.  We have  clarified the  “knowledge-infused” alignment in the paper as it goes beyond plain text-similarity reweighting: ontology features reshape the cross-modal target kernel, so CT embeddings are trained to reflect concept-level clinical structure. This is evidenced by (i) more clinically coherent CT-report similarity blocks in Fig. 3 and (ii) consistent drops in retrieval, generation, and GREEN factuality scores when the knowledge term is removed. We note this component is important, as it provides measurable clinically meaningful gains.
>
> >The experimental evaluation, though extensive, remains narrow in scope and rigor. All tests are CT-based, limiting the claimed cross-domain generalization. Reported gains are modest and lack statistical validation...
>
>  We highlight that our work is primarily scoped to the medical 3D CT domain and is designed around domain-specific characteristics such as volumetric structure, anatomical continuity, and clinical report semantics. While the proposed concept is in principle generalizable to other modalities, this would require domain-aware modifications, which is outside the current paper’s scope. Importantly, our evaluation is not limited to a single data distribution: we train on CT-RATE and perform zero-shot evaluation on BIMCV-R, a fully external dataset with substantially different (1) morphological characteristics, (2) covering different body organs, (3) acquired under different imaging devices and institutional protocols,(4) and has different patient demographics. The table below summarizes the main differences between these two datasets. This setting constitutes a genuine domain shift and therefore a meaningful test of cross-domain generalization.
> | | **CT-RATE** | **BIMCV-R** |
> |---|---|---|
> | **Coverage** | Non-contrast **chest** CT only | **Multi-region/Whole-body** CT |
> | **Hospitals** | Single centre (Istanbul Medipol Univ.) | Multi-centre BIMCV (~11, Valencian Region) |
> | **Vendors** | Philips 61.5%, Siemens 30.1%, PNMS 8.4% | Multiple (not detailed) |
> | **Resolution** | 512, 768, 1024 px | Variable; typically $\geq$512 px |
> | **# of Slices** | 100-600 (avg 305) | 101–670 (avg 279) |
>
> We additionally report the GREEN metric (Ostmeier et al., Findings of ACL: EMNLP 2024), an LLM-based factuality score for radiology reports. GREEN offers: 1) a score aligned with expert preferences, 2) human interpretable explanations of clinically significant errors. SCALE-VLP achieves 0.4054 vs. 0.3955 (M3D), 0.3805 (Merlin), and 0.3721 (CT-CLIP), indicating better clinical correctness.
>
> | Model | GREEN |
> |:-|:-|
> | CT-CLIP | 0.3721 |
> | M3D | 0.3955 |
> | Merlin | 0.3805 |
> | **SCALE-VLP (Ours)** | **0.4054** |
>
>
>
>
> > Overall, the work is a careful system extension rather than a substantive methodological advance. While performance improvements are consistent, the contribution is incremental and lacks the conceptual depth expected for ICLR.
>
> We thank the reviewer for this thoughtful comment and the opportunity to clarify our contribution. While we reuse standard 3D backbones, the primary novelty of this work lies in the novel learning objective. We introduce SWCA, which departs from prior CT-text approaches by replacing identity matrix supervision with structured, instance-adaptive soft targets derived from 3D spatial continuity and knowledge-informed semantic similarity. This reframes CT-text alignment from rigid pairwise matching into distribution alignment that preserves clinically meaningful structure across samples, a concept not present in CT-CLIP, Merlin, fVLM, or M3D. The gains stem from this conceptual advance rather than incremental system tuning.
>
> Furthermore, recent ICLR papers in similar application domains including fVLM (Shui et al., ICLR 2025), which we compared Scale-VLP to it as a baseline, indicate that carefully designed objective-level innovations for CT vision-language modeling are well aligned with the expected depth and scope of ICLR contributions. Our contribution is in the same spirit, but targets the underexplored problem of making the contrastive loss itself volumetric- and semantics-aware.

---

### Official Review · Reviewer_p3ZW · 2025-11-01

**Soundness:** 1
**Presentation:** 3
**Contribution:** 2
**Rating:** 4
**Confidence:** 3

**Summary:**

The paper introduces SCALE-VLP, a dual-encoder framework for 3D CT–report pretraining using a new Soft-Weighted Contrastive Alignment (SWCA) loss. Instead of binary matches, SWCA uses continuous weights that blend spatial coherence in 3D scans (via centroid/covariance kernels) with text similarity from a medical LLM. The setup freezes a RadImageNet 3D ViT and fine-tunes a BioClinicalBERT text encoder, updating only lightweight projection layers. On the CT-RATE benchmark, the model improves retrieval (e.g., +13% R@1), report generation (BLEU-4 0.35, ROUGE-L 0.44), and abnormality classification, with additional zero-shot gains on BIMCV-R. The paper also argues SWCA is more efficient than softmax-based InfoNCE and includes solid ablations for its main components.

**Strengths:**

1. The paper tackles a practical and important issue of how to learn good representations from 3D volumetric scans like CT; this is a step up from the 2D-based work . The authors correctly identify the main issues in this setting: data scarcity , preserving spatial coherence, and the difficulty of integrating complex medical knowledge. The proposed method does address these issues.
2. Consistent multi-task improvements. On CT-RATE: retrieval improves across different pool sizes. The generated reports and classification results seem to improve on all reported metrics. Zero-shot BIMCV-R results are also favorable. The proposed method also shows good generalization.
3. This paper is well-written and generally easy to follow.

**Weaknesses:**

1. Contradiction in the ablation study.  In sec 4.1.1, the authors summarize that "spatial reasoning is the dominant contributor," while removing spatial reasoning results in 2-6% drop but removing both spatial and knowledge results in 14-18% drop. This suggests that knowledge, instead of spatial, accounts for the larger share of the total gain. To the very least the claim that spatial reasoning is the "dominant" is hardly verified. This is evident both in this paragraph and in tab 1 and tab 2.
2. The authors should clarify their main contribution difference vs. prior weighted contrastive losses. The proposed SWCA is close in spirit to SigLIP (pairwise sigmoid) and continuously weighted objectives like CWCL; here weights are built from intra-modal similarity and external priors. The authors should consider a stronger baseline comparison that uses SigLIP/CWCL-style weighting without spatial/knowledge.
3. Although many n-gram related metrics are used for evaluation, this paper does not consider assessment from actual radiologists (human expert), especially for verifying the real-world usefulness of this type of clinically-targeted solutions. The authors should consider doing such eval on a small subset at least.

**Questions:**

Please see weakness.

---

> ### Author Response · Authors · 2025-11-25
> **Author Response (Part 1/2) - Added New Spatial-Only Ablation**
>
> We thank the reviewer for the valuable suggestions and have addressed all concerns as follows (**Part 1/2**). Concerns 2 and 3 are addressed in **Part 2/2**.
>
> > W1. Contradiction in the ablation study. In sec 4.1.1, the authors summarize that "spatial reasoning is the dominant contributor," while removing spatial reasoning results in 2-6% drop but removing both spatial and knowledge results in 14-18% drop. This suggests that knowledge, instead of spatial, accounts for the larger share of the total gain. To the very least the claim that spatial reasoning is the "dominant" is hardly verified. This is evident both in this paragraph and in tab 1 and tab 2.
>
>
>  We conducted an **additional ablation experiment**, and the results are presented in the updated table below; Both priors contribute substantial gains over the no-prior baseline, but the spatial prior consistently yields slightly larger improvements. For retrieval, the spatial-only variant (w/o Medical) improves IR SumR over the base model by +31.0, +18.6, +15.4, and +11.7 for N=100, 500, 1000, and 1564, whereas the knowledge-only variant (w/o Spatial) yields +23.0, +12.6, +14.2, and +11.0, with a similar pattern on TR SumR. **We have updated Table 1 and section 4.4.1  in the paper to reflect the comprehensive version of new ablation**. Also, in Tables 2 and 3 (report generation and abnormality classification), ablating spatial leads to larger drops than ablating knowledge, indicating that spatial coherence is particularly important for downstream tasks. The full SCALE-VLP model, which combines both priors, consistently achieves the best performance.
> We revised Sec. 4.4.1 to state that both spatial and knowledge priors provide sizable, complementary gains over the baseline, and that the largest overall improvements are obtained when they are combined, without claiming either component to be dominant.
>
>
>
> ## $\text{N} = 100$
>
> | Model | IR R@1 | IR R@5 | IR R@10 | IR R@50 | IR R@100 | IR SumR | TR R@1 | TR R@5 | TR R@10 | TR R@50 | TR R@100 | TR SumR |
> |:-|:-|:-|:-|:-|:-|:-|:-|:-|:-|:-|:-|:-|
> | SCALE-VLP w/o Spatial \& Know. | 8.0 | 30.0 | 47.0 | 89.0 | – | 174.0 | 11.0 | 31.0 | 46.0 | 89.0 | – | 177.0 |
> | SCALE-VLP w/o Spatial | 12.0 | 37.0 | 55.0 | 93.0 | – | 197.0 | 13.0 | 40.0 | 54.0 | 92.0 | – | 199.0 |
> | SCALE-VLP w/o Knowledge | **17.0** | 36.0 | **56.0** | **96.0** | – | **205.0** | 10.0 | 37.0 | 58.0 | **94.0** | – | 199.0 |
> | **SCALE-VLP** | 13.0 | **40.0** | **56.0** | 94.0 | – | 203.0 | **14.0** | **42.0** | **59.0** | 93.0 | – | **208.0** |
>
> ---
>
> ## $\text{N} = 500$
>
> | Model | IR R@1 | IR R@5 | IR R@10 | IR R@50 | IR R@100 | IR SumR | TR R@1 | TR R@5 | TR R@10 | TR R@50 | TR R@100 | TR SumR |
> |:-|:-|:-|:-|:-|:-|:-|:-|:-|:-|:-|:-|:-|
> | SCALE-VLP w/o Spatial \& Know. | 3.0 | 8.2 | 14.2 | 46.6 | 66.8 | 138.8 | 2.8 | 9.2 | 15.2 | 45.4 | 64.4 | 137.0 |
> | SCALE-VLP w/o Spatial | 3.2 | 11.6 | 17.8 | 49.4 | 69.4 | 151.4 | 3.2 | 11.2 | 18.8 | 50.4 | 69.0 | 152.6 |
> | SCALE-VLP w/o Knowledge | 2.2 | 11.2 | **20.2** | 52.2 | 71.6 | 157.4 | 3.2 | 10.4 | 18.0 | 51.2 | 71.4 | 154.2 |
> | **SCALE-VLP** | **3.4** | **12.2** | 19.8 | **53.2** | **72.6** | **161.2** | **4.4** | **11.8** | **21.0** | **51.6** | **72.6** | **161.4** |
>
> ---
>
> ## $\text{N} = 1000$
>
> | Model | IR R@1 | IR R@5 | IR R@10 | IR R@50 | IR R@100 | IR SumR | TR R@1 | TR R@5 | TR R@10 | TR R@50 | TR R@100 | TR SumR |
> |:-|:-|:-|:-|:-|:-|:-|:-|:-|:-|:-|:-|:-|
> | SCALE-VLP w/o Spatial \& Know. | 1.5 | 5.0 | 8.5 | 28.3 | 46.0 | 89.3 | 1.3 | 5.5 | 9.2 | 26.9 | 44.6 | 87.5 |
> | SCALE-VLP w/o Spatial | 1.7 | 6.3 | 10.5 | 34.4 | 50.6 | 103.5 | 1.7 | 6.2 | 11.0 | 34.5 | 50.0 | 103.4 |
> | SCALE-VLP w/o Knowledge | 1.1 | 5.3 | 11.0 | 35.2 | **52.1** | 104.7 | 1.8 | 5.8 | 10.0 | 34.1 | **51.8** | 103.5 |
> | **SCALE-VLP** | **1.8** | **6.4** | **11.6** | **35.8** | 51.9 | **107.5** | **2.2** | **6.5** | **11.5** | **35.8** | 51.2 | **107.2** |
>
> ---
>
> ## $\text{N} = 1564$
>
> | Model | IR R@1 | IR R@5 | IR R@10 | IR R@50 | IR R@100 | IR SumR | TR R@1 | TR R@5 | TR R@10 | TR R@50 | TR R@100 | TR SumR |
> |:-|:-|:-|:-|:-|:-|:-|:-|:-|:-|:-|:-|:-|
> | SCALE-VLP w/o Spatial \& Know. | 1.0 | 3.7 | 6.1 | 20.5 | 33.1 | 64.4 | 1.0 | 3.6 | 6.0 | 20.4 | 32.0 | 63.0 |
> | SCALE-VLP w/o Spatial | 1.0 | **4.2** | 7.1 | 24.9 | 38.2 | 75.4 | 1.0 | 4.0 | 7.1 | 24.4 | 38.1 | 74.6 |
> | SCALE-VLP w/o Knowledge | 1.1 | 3.1 | 7.0 | **25.2** | **39.7** | 76.1 | 1.2 | 3.6 | 6.5 | 25.0 | **39.2** | 75.5 |
> | **SCALE-VLP** | **1.2** | **4.2** | **7.4** | 25.1 | 38.8 | **76.7** | **1.3** | **4.2** | **7.4** | **25.4** | 38.5 | **76.8** |

---

> > ### Author Response · Authors · 2025-11-25
> > **Author Response (Part 2/2) - Added New Baseline and Expanded Medical Metrics (GREEN)**
> >
> > > W2. The authors should clarify their main contribution difference vs. prior weighted contrastive losses. The proposed SWCA is close in spirit to SigLIP (pairwise sigmoid) and continuously weighted objectives like CWCL; here weights are built from intra-modal similarity and external priors. The authors should consider a stronger baseline comparison that uses SigLIP/CWCL-style weighting without spatial/knowledge.
> >
> >  We agree that SWCA is, at the objective level, a SigLIP-style pairwise sigmoid loss. Our contribution lies in the construction of soft labels $\(w_{ij} \in [0,1]\)$ for 3D medical CT, derived jointly from (i) intra-modal CT–CT and report–report similarities and (ii) domain-specific spatial and medical-knowledge kernels, so each CT is attracted to clinically related reports and repelled from clearly mismatched ones. CWCL-style losses do not combine both 3D spatial structure and medical priors in this way, nor have they been instantiated for volumetric CT–text alignment. This design is particularly important in radiology, where many “negatives” in CLIP/SigLIP are only partially dissimilar and treating them as equally negative harms the geometry of the embedding space.
> >
> > In line with the reviewer’s suggestion, **we included a CWCL-style as an additional baseline. Full results are in updated Table 1**. This model achieves SumR scores of 152/162 (IR/TR) for N=100, 108.2/110.6 for N=500, 70.1/71.7 for N=1000, and 49.8/51.3 for N=1564, consistently underperforming the SCALE-VLP. This confirms that the gains of SCALE-VLP are not merely due to generic continuous weighting, but stem from the proposed spatial- and knowledge-driven construction of $w_{ij}$.
> >
> > ## $\text{N} = 100$
> >
> > | Model | IR R@1 | IR R@5 | IR R@10 | IR R@50 | IR R@100 | IR SumR | TR R@1 | TR R@5 | TR R@10 | TR R@50 | TR R@100 | TR SumR |
> > |:-|:-|:-|:-|:-|:-|:-|:-|:-|:-|:-|:-|:-|
> > | Baseline (CWCL-style) | 6.0 | 27.0 | 40.0 | 79.0 | – | 152.0 | 9.0 | 29.0 | 44.0 | 80.0 | – | 162.0 |
> > | **SCALE-VLP (Ours)** | **13.0** | **40.0** | **56.0** | **94.0** | – | **203.0** | **14.0** | **42.0** | **59.0** | **93.0** | – | **208.0** |
> >
> > ---
> >
> > ## $\text{N} = 500$
> >
> > | Model | IR R@1 | IR R@5 | IR R@10 | IR R@50 | IR R@100 | IR SumR | TR R@1 | TR R@5 | TR R@10 | TR R@50 | TR R@100 | TR SumR |
> > |:-|:-|:-|:-|:-|:-|:-|:-|:-|:-|:-|:-|:-|
> > | Baseline (CWCL-style) | 2.6 | 6.2 | 12.4 | 34.2 | 52.8 | 108.2 | 2.6 | 7.8 | 11.4 | 35.4 | 53.4 | 110.6 |
> > | **SCALE-VLP (Ours)** | **3.4** | **12.2** | **19.8** | **53.2** | **72.6** | **161.2** | **4.4** | **11.8** | **21.0** | **51.6** | **72.6** | **161.4** |
> >
> > ---
> >
> > ## $\text{N} = 1000$
> >
> > | Model | IR R@1 | IR R@5 | IR R@10 | IR R@50 | IR R@100 | IR SumR | TR R@1 | TR R@5 | TR R@10 | TR R@50 | TR R@100 | TR SumR |
> > |:-|:-|:-|:-|:-|:-|:-|:-|:-|:-|:-|:-|:-|
> > | Baseline (CWCL-style) | 1.1 | 4.1 | 6.9 | 23.0 | 35.0 | 70.1 | 1.1 | 4.7 | 7.4 | 23.1 | 35.4 | 71.7 |
> > | **SCALE-VLP (Ours)** | **1.8** | **6.4** | **11.6** | **35.8** | **51.9** | **107.5** | **2.2** | **6.5** | **11.5** | **35.8** | **51.2** | **107.2** |
> >
> > ---
> >
> > ## $\text{N} = 1564$
> >
> > | Model | IR R@1 | IR R@5 | IR R@10 | IR R@50 | IR R@100 | IR SumR | TR R@1 | TR R@5 | TR R@10 | TR R@50 | TR R@100 | TR SumR |
> > |:---|:---|:---|:---|:---|:---|:---|:---|:---|:---|:---|:---|:---|
> > | Baseline (CWCL-style) | 0.6 | 3.1 | 4.5 | 16.1 | 25.5 | 49.8 | 0.6 | 3.0 | 5.0 | 17.1 | 25.6 | 51.3 |
> > | **SCALE-VLP (Ours)** | **1.2** | **4.2** | **7.4** | **25.1** | **38.8** | **76.7** | **1.3** | **4.2** | **7.4** | **25.4** | **38.5** | **76.8** |
> >
> > > W3. Although many n-gram related metrics are used for evaluation, this paper does not consider assessment from actual radiologists (human expert), especially for verifying the real-world usefulness of this type of clinically-targeted solutions. The authors should consider doing such eval on a small subset at least.
> >
> >  In the revised version, **we additionally report the GREEN metric (Ostmeier et al., Findings of ACL: EMNLP 2024)**, a recent LLM-based evaluator for radiology report factuality, both below and in Table 2. GREEN offers: 1) a score aligned with expert preferences, 2) human interpretable explanations of clinically significant errors. Using the official implementation, SCALE-VLP achieves a GREEN score of 0.4054, outperforming other baselines, indicating stronger clinical correctness beyond n-gram overlap. We agree that expert human evaluation is important; however, consistent with prior CT-RATE/BIMCV-R work, we focus on automatic but clinically grounded metrics (retrieval with real reports, multi-label abnormality classification, BERT-F1/CIDEr, and GREEN). Conducting a properly powered radiologist reader study requires IRB approval and coordination, so we explicitly state this as a limitation and propose as future work a blinded study where radiologists assess factual accuracy.
> > | Model | GREEN |
> > |:-|:-|
> > | CT-CLIP | 0.3721 |
> > | M3D | 0.3955 |
> > | Merlin | 0.3805 |
> > | **SCALE-VLP (Ours)** | **0.4054** |

---

### Author Response · Authors · 2025-11-27
**General Response and Key Improvements**

We sincerely appreciate the thoughtful feedback from Reviewers p3ZW, GLiq, BXEP, and TmJw, as well as their constructive suggestions on how to improve the paper. We are encouraged that reviewers found our problem setting **important and underexplored** in 3D CT vision–language pretraining (p3ZW, GLiq, BXEP), viewed SCALE-VLP as a **novel and promising** extension of soft contrastive learning for CT VLMs (GLiq, BXEP), appreciated that it provides **a single pretraining framework with a shared encoder** for three clinically relevant downstream tasks (Tmjw), and highlighted our **consistent multi-task gains** on CT-RATE (retrieval, report generation, and abnormality classification across pool sizes) and **favorable zero-shot generalization** to BIMCV-R (p3ZW, GLiq, BXEP, Tmjw) . Reviewers also noted that our paper is **well-structured and clearly motivated** (p3ZW, GLiq), that the soft-weighted alignment idea is **reasonable and clearly explained** (GLiq), and that our evaluation provides **detailed quantitative results** across retrieval, report generation, and abnormality classification on both in-domain and out-of-domain CT datasets (BXEP, TmJw).


In this rebuttal, we address each concern in detail and, guided by the feedback, we:

* **Expanded Ablations and Baselines:**

Provide clearer ablations that separate (i) SWCA with intra-modal (CWCL-style) weights only, (ii) SWCA + spatial prior, (iii) SWCA + knowledge weighting, and (iv) full SWCA + spatial + knowledge weighting. These results show that both spatial and knowledge priors contribute, and that SCALE-VLP’s gains go beyond generic continuous reweighting.

* **Knowledge Component & External Medical LLMs:**

Evaluate SCALE-VLP with three public medical LLMs (HuatuoGPT, LLaMA3-Med42, BioMistral) to show that performance is stable across backbones and that the framework uses the LLM as a modular knowledge prior rather than depending on a particular model.

* **Clinical Correctness:**

Complement n-gram metrics with the GREEN metric for radiology reports, which offers: i) a score aligned with expert preferences, ii) human interpretable explanations of clinically significant errors. The results also demonstrate SCALE-VLP outperforms prior methods.

* **Clarified Novelty and Relation to CLIP/SigLIP/CWCL:**

Sharpen the positioning that our contribution is at the objective level: replacing identity-style supervision with structured, instance-adaptive soft targets that encode 3D spatial continuity and medical knowledge, a design absent in CT-CLIP, Merlin, fVLM, and M3D.

* **Improved Clarity and Presentation:**

Refine dataset descriptions, table legends, and terminology (e.g., “knowledge-informed weighting” instead of “fusion”), in order to make the paper more accessible to researchers across machine learning and medical imaging.

&nbsp;



We believe these clarifications and additional experiments strengthen the paper’s technical contribution and empirical support, and we are grateful to the reviewers for helping us improve the work.

Kindly consider adjusting our overall score if our response addressed your primary concerns. We would be happy to answer any additional questions you may have in order for you to support acceptance of our work.

---

### Note · Authors · 2026-01-06

**Comment:**

We thank the reviewers for their constructive and helpful feedback, which has been valuable for revising the paper. However, we have decided to withdraw this submission due to recent changes in the review process.

**Withdrawal Confirmation:**

I have read and agree with the venue's withdrawal policy on behalf of myself and my co-authors.